# Thompson Sampling For Combinatorial Bandits: Polynomial Regret and Mismatched Sampling Paradox

**Raymond Zhang**
Laboratoire des signaux et systèmes
Université Paris-Saclay, CNRS, CentraleSupélec,
91190, Gif-sur-Yvette, France.
`Raymond.zhang@centralesupelec.fr`

**Richard Combes**
Laboratoire des signaux et systèmes
Université Paris-Saclay, CNRS, CentraleSupélec,
91190, Gif-sur-Yvette, France.
`Richard.combes@centralesupelec.fr`

## Abstract

We consider Thompson Sampling (TS) for linear combinatorial semi-bandits and subgaussian rewards. We propose the first known TS whose finite-time regret does not scale exponentially with the dimension of the problem. We further show the mismatched sampling paradox: A learner who knows the rewards distributions and samples from the correct posterior distribution can perform exponentially worse than a learner who does not know the rewards and simply samples from a well-chosen Gaussian posterior. The code used to generate the experiments is available at https://github.com/RaymZhang/CTS-Mismatched-Paradox

## 1 Introduction and Setting

We consider the linear combinatorial bandit problem with semi-bandit feedback: At time $t \in [T] := \{1, ..., T\}$ a learner selects an action $A(t) \in \mathcal{A}$ where the set of available actions $\mathcal{A} \subset \{0, 1\}^d$ is a known combinatorial set, i.e., a set of binary vectors. Then the environment draws a random vector $X(t) \in \mathbb{R}^d$, and the learner then observes $Y(t) = A(t) \odot X(t)$, where $\odot$ denotes the Hadamard (element-wise) product, and the learner obtains a reward of $A(t)^\top X(t)$. We assume that the vectors $(X(t))_{t \geqslant 1}$ are drawn i.i.d. from some distribution with expectation $\mathbb{E}(X(t)) = \mu^\star$ and that the entries of $X(t)$ are independent. The vector $\mu^\star$ lies in some $\Theta \subset \mathbb{R}^d$ and is initially unknown to the learner. The learner wants to minimize the regret:

$$R(T, \mu^\star) := T \max_{A \in \mathcal{A}} \left\{ A^\top \mu^\star \right\} - \mathbb{E}\left[ \sum_{t \in [T]} A(t)^\top \mu^\star \right] = \mathbb{E}\left[ \sum_{t \in [T]} \Delta_{A(t)} \right].$$

Where $A^\star \in \arg\max_{A \in \mathcal{A}} \{A^\top \mu^\star\}$ is the optimal action and $\Delta_A := A^{\star\top} \mu^\star - A^\top \mu^\star$ is the reward gap between action $A$ and optimal action $A^\star$. The regret is the expected difference between the sum of rewards obtained by an oracle who knows $\mu^\star$ and always selects the optimal decision and that obtained by the learner. We assume that the optimal decision is unique. To state regret bounds, we use the following notation. We denote by $\Delta_{\min} := \min_{A \in \mathcal{A}: \Delta_A > 0} \Delta_A$ the minimal reward gap and $\Delta_{\max} := \max_{A \in \mathcal{A}} \Delta_A$ the maximal reward gap, $\Delta(t) := \Delta_{A(t)}$ the reward gap of the action selected at time $t$, $m := \max_{A \in \mathcal{A}} \|A\|_1$ the maximal size of an action.

38th Conference on Neural Information Processing Systems (NeurIPS 2024).

The regret depends on the distributions of the random vector $X(t)$, which generates the rewards, and we will assume throughout the paper that $X(t)$ is $\sigma^2$-subgaussian so that for all $\lambda \in \mathbb{R}^d$:

$$\mathbb{E}[\exp(\lambda^\top X(t))] \leqslant e^{\lambda^\top \mu^\star + \frac{\|\lambda\|_2^2 \sigma^2}{2}}.$$

This assumption holds in many scenarios of interest, for instance, when $X(t) \in [a,b]^d$ with $\sigma^2 = (b-a)^2/4$, or when $X(t)$ is normally distributed with a covariance matrix smaller than $\sigma^2 I_d$. We assume that $\sigma$ is known, or at least upper-bounded.

One of the candidate algorithms for this problem is Thompson Sampling (TS), which is based on Bayesian inference. We consider a prior distribution $\pi_0$ over $\Theta$, a likelihood function $\ell$ and $\pi_t$ the *posterior distribution* of $\mu^\star$ at time $t$ knowing the observations and the actions up to time $t$:

$$\pi_t(\mu) = \frac{\prod_{s \in [t-1]} \prod_{i \in [d]} [A_i(s)\ell(X_i(s), \mu_i) + (1 - A_i(s))]\pi_0(\mu)}{\int_\Theta \prod_{s \in [t-1]} \prod_{i \in [d]} [A_i(s)\ell(X_i(s), \mu_i) + (1 - A_i(s))]\pi_0(\mu)d\mu}$$

The TS algorithm with prior $\pi_0$ and likelihood $\ell$ consists in sampling $\theta(t)$ from distribution $\pi_t$, and select action $A(t) \in \arg\max_{a \in \mathcal{A}} A^\top \theta(t)$. The random vector $\theta(t)$, called a Thompson Sample, acts as a proxy for the unknown $\mu^\star$ and guides the exploration.

We will show how TS behaves with either Gaussian likelihood $\ell(x, \mu) = (2\pi)^{-1/2} e^{-(x-\mu)^2/2}$ or Bernoulli likelihood $\ell(x, \mu) = \mathbf{1}\{x = 1\}\mu + (1 - \mu)\mathbf{1}\{x = 0\}$ in the next sections. If the likelihood $\ell(X_i(s), \mu_i)$ is selected as the distribution of $X_i(s)$ knowing $\mu_i$ then we say that TS is natural. If the likelihood $\ell(X_i(s), \mu_i)$ is different from the distribution of $X_i(s)$ knowing $\mu_i$, we will say that TS is mismatched. An example of mismatched TS would be to select $\ell$ as the Gaussian likelihood, although the actual distribution of $X(t)$ knowing $\mu^\star$ is, say, Bernoulli or some other bounded distribution. Ultimately, the likelihood function is a choice left up to the learner to control how TS explores the suboptimal actions. Perhaps counterintuitively, mismatched TS can outperform natural TS, as we will demonstrate. The prior $\pi_0$ can be chosen in various ways, for instance, Jeffrey's non-informative prior. It can even be chosen as an improper prior, where $\int_\Theta \pi_0(\mu)d\mu = +\infty$, as long as the integral in the definition of $\pi_t$ is well-defined.

TS is usually computationally simple to implement as it requires a linear maximization over the action space $\mathcal{A}$ at each step, which we assume can be done in polynomial time in $m$ and $d$. This fact explains the practical appeal of TS since whenever linear maximization over $\mathcal{A}$ can be implemented efficiently; the algorithm has low computational complexity. Also, for some problem instances, it tends to perform well numerically.

## 2 Related Work and Contribution

Combinatorial bandits are a generalization of classical bandits studied in [17]. Several asymptotically optimal algorithms are known for classical bandits, including the algorithm of [18], KL-UCB [5], DMED [13] and TS [14, 22]. Other algorithms include the celebrated UCB1 [4]. Numerous algorithms for combinatorial semi-bandits have been proposed, many of which naturally extend algorithms for classical bandits to the combinatorial setting. CUCB [6, 16] is a natural extension of UCB1 to the combinatorial setting. ESCB [8, 10] is an improvement of CUCB, which leverages the independence of rewards between items. AESCB [9] is an approximate version of ESCB with roughly the same performance guarantees and reduced computational complexity. TS for combinatorial semi bandits was considered in [11, 15, 20, 21, 23] while TS for linear bandit was studied in [1, 3]. Also, combinatorial semi-bandits are a particular case of structured bandits, for which there exist asymptotically optimal algorithms such as OSSB [7]. It was also shown in [15] that TS is asymptotically and finite time optimal for matroid-like action sets. The Bayesian regret of TS has been extensively studied, e.g., [21].

Three types of regret bounds exist in the literature: $R(T, \mu^\star)$ is the problem-dependent regret, i.e., the regret of the learner on the particular problem instance defined by $\mu^\star$. The minimax regret is $\max_{\mu \in \Theta} R(T, \mu^\star)$ and the Bayesian regret $\mathbb{E}_{\mu^\star}[R(T, \mu^\star)]$ where $\mu^\star$ is drawn according to some prior distribution. We will state regret bounds as a function of the parameters $T$, the time horizon; $d$, the problem dimension; $m$, the maximal size of an action; and $\Delta_{\min}$, the minimal gap.

The paper [16] showed that the problem-dependent regret of CUCB is upper bounded by $O\left(\frac{dm \ln T}{\Delta_{min}}\right)$ and its minimax regret is upper bounded by $O(\sqrt{\sigma^2 dmT \ln T} + dm)$. [10] showed that the problem-

dependent regret of ESCB is upper bounded by $O\left(\frac{\sigma^2 d(\ln m)^2 \ln T}{\Delta_{min}} + \frac{dm^3}{\Delta_{min}^2}\right)$ and its minimax regret is upper bounded by $O(\sqrt{d(\ln m)^2 T \ln T} + dm)$. [20] showed that the problem-dependent regret of TS is upper bounded by

$$O\left(\frac{\sigma^2 d(\ln m)^2}{\Delta_{\min}} \ln T + \frac{dm^3}{\Delta_{\min}^2} + m\left(\sigma \frac{m^2+1}{\Delta_{min}}\right)^{2+4m}\right).$$

While this bound is almost optimal in the asymptotic regime where $T \to \infty$, it is exponentially suboptimal in the finite time regime since the last term of this expression scales exponentially with $m$. Unless one assumes a particular type of action set as in [15], all the known generic regret upper bound for TS in the literature [11], [23], [20] feature an exponential dependency on $m$. [24] further showed that the problem-dependent regret of TS for some simple combinatorial set can be lower bounded by an expression scaling exponentially in $m$,

$$R(T, \mu^\star) \geqslant \frac{\Delta_{\min}}{4 p_{\Delta_{\min}}}(1-(1-p_{\Delta_{\min}})^{T-1}), \text{ with } p_{\Delta_{\min}} = \exp\left\{-\frac{2m}{9}\left(\frac{1}{2} - (\frac{\Delta_{\min}}{m} + \frac{1}{\sqrt{m}})\right)^2\right\}.$$

They further showed that Thompson Sampling is not minimax optimal for some combinatorial bandit problems. Therefore, the exponential term is not an artefact of the analysis of [20]. It is also noted that the regret upper bounds for ESCB and CUCB do not feature this exponential dependency in $m$, suggesting that those algorithms are better in the finite time regime than the versions of TS analyzed so far in the literature. This is unfortunate because TS usually has very low computational complexity, and having an algorithm with both low computational complexity and low regret in the finite time regime would be highly desirable.

**Our contribution : (i)** We propose a new variant of TS with a regret upper bounded by:

$$O\left(\frac{\sigma^2 d \ln m}{\Delta_{\min}} \ln T + \frac{\sigma^2 d^2 m \ln m}{\Delta_{\min}} \ln\ln T + P\left(m, d, \frac{1}{\Delta_{\min}}, \Delta_{\max}, \sigma\right)\right)$$

where $P$ is a polynomial in $m, d, \frac{1}{\Delta_{\min}}, \Delta_{\max}$. This polynomial term is a clear improvement over the bound of [20] in the finite time, high dimensional regime where $T$ is relatively small and $m$ is large. Indeed, the last term in this bound $P\left(m, d, 1/\Delta_{\min}, \Delta_{\max}, \sigma\right)$ will be much smaller than the last term in the bound of [20] $m\left(\sigma(m^2+1)/\Delta_{\min}\right)^{2+4m}$ which is exponential in $m$. To design our variant, we add a slight exploration boost to TS, which vanishes as $T \to \infty$ but significantly impacts the algorithm behaviour when $T$ is moderate and $m$ is large. Also note that the improvement in the $\ln m$ term comes from a direct application of a result in [19].

**(ii)** We design new proof strategies to derive this upper bound, which are based on carefully bounding the sample path behaviour of TS. We believe those strategies are an essential contribution to the analysis of TS and enable us to show that with high probability, TS will sample the optimal action at least $\Omega(t^\alpha)$ times with $\alpha > 0$. This number serves to control the transient behaviour of TS.

**(iii)** As a by-product, we show the mismatched sampling paradox of TS: in some cases, mismatched TS performs exponentially better than natural TS. For instance, in a problem where $X(t)$ has Bernoulli distribution, a learner using a uniform (improper) prior and a Gaussian likelihood can perform exponentially better than a learner using a Beta prior (which includes Jeffreys' prior) and the Bernoulli likelihood. In essence, trying to exploit the learner's statistical knowledge about the model ends up harming them.

**(iv)** We confirm our theoretical predictions using numerical experiments, which clearly show that our variant of TS outperforms by several orders of magnitude the Beta-based versions studied in the literature whose regret scales exponentially in the ambient dimension.

## 3   Algorithms

In this section, we present three TS algorithms: B-CTS (Beta-Combinatorial Thompson Sampling), which is TS with a beta prior and a Bernoulli likelihood, G-CTS (Gaussian-Combinatorial Thompson Sampling) which is TS with a uniform (improper) prior and a Gaussian likelihood, and finally BG-CTS (Boosted Gaussian-Combinatorial Thompson Sampling), an algorithm we propose by introducing a carefully chosen exploration boost in G-CTS.

## 3.1 Notation

We use the following notation to state algorithms. We define the statistics $N(t) := \sum_{s \in [t-1]} A(s)$, the vector containing the number of times each item has been selected, $M_A(t) := \sum_{s \in [t-1]} \mathbf{1}\{A(s) = A\}$ the number of times action $A$ has been selected until $t$, $V(t) := D_{N(t)}^{-1}$ the diagonal matrix whose diagonal elements are $(1/N_1(t), ..., 1/N_d(t))$, $\widehat{\mu}(t) := V(t) \sum_{s \in [t-1]} X_i(s) \odot A_i(s)$ the empirical average estimator for $\mu^\star$. We denote by $\mathcal{H}(t) = (A(s), A(s) \odot X(s))_{s \in [t-1]}$ the history which contains all the information collected by the learner up to time $t$, and which includes both the observations and the selected actions. For two vectors $\alpha, \beta$ in $(\mathbb{R}^+)^d$ we denote by $\text{Beta}(\alpha, \beta) = \bigotimes_{i=1}^d \text{Beta}(\alpha_i, \beta_i)$ the distribution of a vector with independent entries, and where the $i$-th entry is $\text{Beta}(\alpha_i, \beta_i)$ distributed.

## 3.2 B-CTS

B-CTS (see 1 in the algorithm format) considers the prior $\pi_0 = \text{Beta}(\alpha(0), \beta(0))$ where $\alpha(0), \beta(0)$, are two vectors in $\mathbb{R}^d$ chosen by the learner and the Bernoulli likelihood $\ell(x, \mu) = \mathbf{1}\{x = 1\}\mu + (1 - \mu)\mathbf{1}\{x = 0\}$. If $\alpha(0) = \beta(0) = (1, ..., 1)$, then the prior $\pi_0$ is uniform over $[0, 1]^d$. If $\alpha(0) = \beta(0) = (1/2, ..., 1/2)$, then the prior is Jeffreys' non-informative prior, which is proportional to the square root of the determinant of the Fisher information matrix. In B-CTS, the posterior distribution $\pi_t$ is also a Beta distribution so that the Beta-CTS selects the action:

$$A(t) \in \arg\max_{A \in \mathcal{A}} A^\top \theta(t) \text{ with } \theta(t) \sim \text{Beta}(\alpha(t), \beta(t))$$

where vectors $\alpha(t), \beta(t)$ are defined as:

$$\alpha(t) := \sum_{s \in [t-1]} X(s) \odot A(s) + \alpha(0) \text{ and } \beta(t) := \sum_{s \in [t-1]} (1 - X(s)) \odot A(s) + \beta(0).$$

## 3.3 G-CTS

G-CTS considers the improper prior $\pi_0$ which is constant and equal to $1/\sigma$ on all $\mathbb{R}^d$ and Gaussian likelihood $\ell(x, \mu) = (2\pi\sigma^2)^{-1/2} e^{-(X-\mu)^2/(2\sigma^2)}$, where $\sigma^2$ is the variance. Of course, since $\pi_0$ is improper, for $\pi_t$ to be well-defined, we require that enough samples have been collected so that $N(t) \geqslant 1$. This is easily achieved by selecting $d$ actions $A^1, ..., A^d$ that cover $\mathcal{A}$ in the sense that $\sum_{i \in [d]} A^i \geqslant 1$, and initializing the algorithm by sampling each of them once. In G-CTS, the posterior distribution $\pi_t$ is also a Gaussian distribution so that the G-CTS selects the action:

$$A(t) \in \arg\max_{A \in \mathcal{A}} A^\top \theta(t) \text{ with } \theta(t) \sim N(\widehat{\mu}(t), \sigma^2 V(t)).$$

## 3.4 BG-CTS

BG-CTS (see 2 in the algorithm format) is a modification of G-CTS that we propose and that selects the action

$$A(t) \in \arg\max_{A \in \mathcal{A}} A^\top \theta(t) \text{ with } \theta(t) \sim N(\widehat{\mu}(t), 2g(t)\sigma^2 V(t)) \text{ with}$$

$$g(t) := \frac{f(t)}{\ln t} \text{ and } f(t) := (1 + \lambda)\left(\ln t + (m + 2)\ln\ln t + \frac{m}{2}\ln\left(1 + \frac{e}{\lambda}\right)\right)$$

and $\lambda \in \mathbb{R}^+$ is an input parameter of the algorithm. BG-CTS behaves like G-CTS with a time-varying boost in its exploration denoted by $g(t)$. This boost asymptotically behaves like a constant $\lim_{t \to \infty} g(t) = 1 + \lambda$. This boost ensures a much better finite-time behaviour, especially in the moderate $T$, large $m$ regime, to avoid the exponentially large regret that can occur in TS. The form of $g(t)$ is not arbitrary and is derived from the self-normalized concentration inequalities that control the large deviations of vector $\widehat{\mu}(t)$. To make our analysis clearer, we will assume that there exists an exogenous process $(Z(t))_{t \geqslant 1}$ of i.i.d. $\mathcal{N}(0, I_d)$ vectors that serves as the random generator number for the Thompson samples with $\theta(t) = \widehat{\mu}(t) + \sigma\sqrt{2g(t)}V^{\frac{1}{2}}(t)Z(t)$.

This decomposition is useful for separating the algorithm's randomness from the bandit environment's randomness. We notice that for all $s \geqslant t$, $Z(s)$ and the history $\mathcal{H}(t)$ are independent. Furthermore, we call $Z(s)$ the random part of the Thompson sample, $A^\top \theta(t)$ the Thompson sample of action $A$

## 4  Main Result

We now state Theorem 1, our main result.

**Theorem 1.** *For $\lambda = 1$, and $\sigma^2$ subgaussian rewards, the regret of BG-CTS is upper bounded by:*

$$R(T, \mu^\star) \leqslant C \frac{\sigma^2 d \ln m}{\Delta_{\min}} \ln T + C' \frac{\sigma^2 d^2 m \ln m}{\Delta_{\min}} \ln \ln T + P\left(m, d, \frac{1}{\Delta_{\min}}, \Delta_{\max}, \sigma\right) \quad (1)$$

*with $C, C'$ universal constants and $P$ a polynomial in $m, d, \frac{1}{\Delta_{\min}}, \Delta_{\max}, \sigma$.*

Theorem 1 states that the regret of BG-CTS is upper bounded by an expression with both the correct behaviour when $T$ is large i.e., both this bound and that of [20] give the same upper bound on $\limsup_{T\to\infty} \frac{R(T,\mu^\star)}{\ln T}$, but also a polynomial dependency in $m, d, \frac{1}{\Delta_{\min}}, \Delta_{\max}, \sigma$. This result predicts that BG-CTS performs much better than other TS variants in the regime where the time horizon $T$ is moderate and the decision size $m$ is large.

A consequence of Theorem 1 combined with prior known results of [24] this is the mismatched sampling paradox for TS: a learner attempting to leverage his knowledge about the statistical model by using natural TS can perform exponentially worse than a learner willingly ignoring this knowledge and using mismatched TS by using an algorithm such as BG-CTS. Consider the example of [24] which features two disjoint actions of size $m = d/2$ written $(1, ..., 1, 0, ..., 0)$ and $(0, ..., 0, 1, ..., 1)$ and Bernoulli rewards. Suppose the learner attempts to leverage that she knows the rewards are Bernoulli and that the parameter space is $[0, 1]^d$. She will employ a uniform prior over $[0, 1]^d$ and the Bernoulli likelihood. This means using B-CTS and getting a regret that scales exponentially with $d$ as shown in [24]. Using B-CTS with Jeffrey's prior does not help either. On the other hand, if the learner pretends she does not know the parameter space nor the rewards distribution and uses B-CTS, she gets a regret scaling only polynomially in $d$. Furthermore, Bernoulli rewards are $\sigma^2$ subgaussian with $\sigma^2 \leqslant 1/4$ as stated above, so our regret upper bound for BG-CTS applies to this example.

At first glance, it seems outright absurd to use a prior whose support is the whole of $\mathbb{R}^d$ instead of the actual parameter space $[0, 1]^d$, and using a Gaussian likelihood, which is continuous when the rewards are binary, but this paradoxically gives exponentially better performance. This paradox leads us to believe one should be careful when using posterior sampling for regret minimization. While this is natural for Bayesian inference, things seem to be much more complex when solving bandit problems, which feature both inference and control/exploration.

## 5  Regret Analysis

In this section, we describe how to prove our main result. Due to space constraints, some proof elements are relegated to the appendix. In particular, to make this proof self-contained, we reproduce (without their proofs) the results from previous work that we use for our analysis. A reader can try to follow the proof with the help of the diagram in figure 2.

A fundamental idea of our analysis is to consider the event $\mathfrak{A}_t$ where both events occur :

$$\forall s \in [t], \forall A \in \mathcal{A} : |A^\top \theta(t) - A^\top \mu^\star| \leqslant C_1 \sigma \sqrt{m \ln t} A^\top V^{\frac{1}{2}}(s) A,$$

$$\text{and } |\{s \in [t] : A^{\star\top} \theta(s) \geqslant A^{\star\top} \mu^\star\}| \geqslant C_2 t^\alpha$$

Where $C_1 = \sqrt{8} + \sqrt{72}$, $C_2 = \frac{1}{2^{13/4 + C_3^2/2} C_3 \sqrt{2\pi}}$, $(C_3)^2 = 1.238$, $\alpha = 3/4 - (1/2)(C_3)^2 \approx 0.131$.

When $\mathfrak{A}_t$ occurs, we say that we observe a clean run up to time $t$. A clean run up to time $t$ implies that the Thompson sample of any action $A$ at any time $s \in [t]$ cannot exceed the sum of its expected value and a bonus proportional to $A^\top V^{\frac{1}{2}}(t) A$, which can be interpreted as the confidence bonus used in the CUCB algorithm. A clean run also implies that there exist many instants at which the Thompson sample of the optimal action is at least as large as its expected reward $A^{\star\top} \mu^\star$.

### 5.1  Probability of observing a clean run

We first now show that most runs are clean, i.e., clean runs occur with high probability. Proposition 2 states that the probability of a non-clean run up to time $t$ is much smaller than $1/t$, and therefore non-clean runs cause little regret.

**Proposition 2.** *For all $t \geqslant C_5$, we have $\mathbb{P}(\mathfrak{A}_t) \geqslant 1 - 4dt^{-2} - t^{-1}(\ln t)^{-2} - e^{-C_4 t^\alpha}$ with $C_4 = C_2/8$ and $C_5 = 23$.*

**Proof :** The proof is relatively technical, and involves decomposing $\mathfrak{A}_t$ according to the fluctuations of $\theta(t)$ and $\widehat{\mu}(t)$. We decompose the Thompson sample of the optimal action as follows:

$$A^{\star\top}\theta(s) = A^{\star\top}\mu^\star + [U^\star(s) + S^\star(s)]\sqrt{A^{\star\top}V(s)A^\star}$$

with

$$U^\star(s) := \frac{A^{\star\top}(\widehat{\mu}(s) - \mu^\star)}{\sqrt{A^{\star\top}V(s)A^\star}} \text{ and } S^\star(s) := \sigma\sqrt{2g(s)}\frac{A^{\star\top}V^{1/2}(s)Z(s)}{\sqrt{A^{\star\top}V(s)A^\star}}$$

which represent the deviation between the empirical mean and the expected reward, and the deviation of the Thompson sample from the expected value of the Thompson sample.

We introduce the following deviation events

$$\mathfrak{B}_t := \{\max_{s\in[t]} \|V^{-\frac{1}{2}}(s)(\mu^\star - \widehat{\mu}(s))\|_\infty \geqslant \sigma\sqrt{8\ln t}\} \qquad \mathfrak{C}_t := \{\max_{s\in[t]} \|Z(s)\|_\infty \geqslant \sqrt{6\ln t}\}$$

$$\mathfrak{E}_t := \{|\{s \in [t] : S^\star(s) \geqslant \sigma\sqrt{2f(t)}\}| \leqslant C_2 t^\alpha\} \qquad \mathfrak{D}_t := \{\max_{s\in[t]} U^\star(s) \geqslant \sigma\sqrt{2f(t)}\}.$$

Each of those events can be interpreted as follows. $\mathfrak{B}_t$ means that the empirical mean of some item deviates from its expected value at least once, $\mathfrak{C}_t$ means that the randomization in the Thompson sample is abnormally large at least once, $\mathfrak{D}_t$ implies that the empirical mean of the optimal action deviates from its expected value at least once, and $\mathfrak{E}_t$ means that there exist too few instants at which $S^\star(t)$ is reasonably large.

Assume that none of $\mathfrak{B}_t$, $\mathfrak{C}_t$, $\mathfrak{D}_t$, $\mathfrak{E}_t$ occur. For all $A \in \mathcal{A}$ and all $s \in [t]$:

$$|A^\top\theta(s) - A^\top\mu^\star| \leqslant |A^\top(\widehat{\mu}(s) - \mu^\star)| + |A^\top(\theta(s) - \widehat{\mu}(s))| \leqslant C_1\sigma\sqrt{m\ln t}A^\top V^{1/2}(s)A$$

since if $\mathfrak{B}_t$ does not occur:

$$|A^\top(\widehat{\mu}(s) - \mu^\star)| \leqslant \sigma\sqrt{8\ln t}A^\top V^{1/2}(s)A$$

and if $\mathfrak{C}_t$ does not occur and because $g(t) < 2(2m + 1) < 6m$ see lemma f. 7:

$$|A^\top(\theta(s) - \widehat{\mu}(s))| \leqslant \sigma\sqrt{72m\ln t}A^\top V^{1/2}(s)A$$

Furthermore if $\mathfrak{D}_t$ and $\mathfrak{E}_t$ do not occur, there exists at least $C_2 t^\alpha$ instants such that $S^\star(s) \geqslant \sigma\sqrt{2f(t)}$ and $U^\star(s) \geqslant -\sigma\sqrt{2f(t)}$, which implies $A^{\star\top}\theta(s) \geqslant A^{\star\top}\mu^\star$. This means that:

$$|\{s \in [t] : A^{\star\top}\theta(t) \geqslant A^{\star\top}\mu^\star\}| \geqslant C_2 t^\alpha$$

Therefore $\mathfrak{A}_t$ occurs, and we have a clean run. Hence $\mathbb{P}(\mathfrak{A}_t) \geqslant 1 - \mathbb{P}(\mathfrak{B}_t) - \mathbb{P}(\mathfrak{C}_t) - \mathbb{P}(\mathfrak{D}_t) - \mathbb{P}(\mathfrak{E}_t)$.

We now upper bound the probability of each event separately.

### 5.1.1  Probability of $\mathfrak{B}_t$

Using a union bound $\mathbb{P}(\mathfrak{B}_t) \leqslant \sum_{i\in[d]} \mathbb{P}\left(\max_{s\in[t]} \sqrt{V_i(s)}(\mu_i^\star(s) - \widehat{\mu}(s)) \geqslant \sigma\sqrt{8\ln t}\right) \leqslant 2dt^{-2}$. We used the concentration inequality first derived by [16] in their analysis of CUCB and recalled in lemma 7.

### 5.1.2  Probability of $\mathfrak{C}_t$

Using a union bound and a Chernoff bound for the Gaussian distribution (lemma 10), wheres $Q$ is the tail function of the standard Gaussian distribution :

$$\mathbb{P}(\mathfrak{C}_t) \leqslant \sum_{i\in[d]} \sum_{s\in[t]} \mathbb{P}\left(|Z_i(s)| \geqslant \sqrt{6\ln t}\right) \leqslant 2tdQ(\sqrt{6\ln t}) \leqslant 2td\exp(-3\ln t) = 2dt^{-2}$$

### 5.1.3 Probability of $\mathfrak{D}_t$

We have for $t \geqslant 2, \mathbb{P}(\mathfrak{D}_t) \leqslant t^{-1}(\ln t)^{-2}$ from the concentration inequality derived by [10] in their analysis of ESCB and recalled in lemma 5.

### 5.1.4 Probability of $\mathfrak{E}_t$

In order to control the probability of $\mathfrak{E}_t$, consider the following counting process:

$$W(s) = \sum_{u \in [s]} \mathbf{1}\{S^\star(u) \geqslant \sigma\sqrt{2f(t)}\}$$

We wish to show that, with high probability, $W(s) \geqslant C_2 t^\alpha$. One may readily check that $W(s)$ is a sum of binary variables and that its conditional expected increment verifies:

$$p(s) = \mathbb{E}(W(s) - W(s-1)|\mathcal{H}(s)) = \mathbb{P}(S^\star(s) \geqslant \sigma\sqrt{2f(t)}|\mathcal{H}(s)) = Q(\sqrt{f(t)/g(s)})$$

since, conditional to $\mathcal{H}(s)$, $S^\star(s)$ has a gaussian distribution with mean 0 and variance $2g(s)$. Let us lower bound of the sum of $p$. By considering $t \geqslant C_5$ we have

$$\sum_{s \in [t]} p(s) \geqslant (t/2)p(t/2) = (t/2)Q(\sqrt{f(t)/g(t/2)}) \geqslant (t/2)Q(C_3\sqrt{\ln(t/2)}) \geqslant 2C_2 t^\alpha$$

using the fact that $p$ is increasing in $s$, and the study of $\frac{f(t)}{g(t/2)}$ done in lemma f. 1 , and lemma 10 on the asymptotic behaviour of the $Q$ function.

We can now conclude by applying a multiplicative Azuma-Hoeffding style bound to $W(s)$ presented in lemma 6 in the appendix. With $C_4 = C_2/8$ we have :

$$\mathbb{P}(\mathfrak{E}_t) \leqslant \mathbb{P}\left(W(t) \leqslant C_2 t^\alpha\right) \leqslant \mathbb{P}\left(W(t) \leqslant (1/2)\sum_{s \in [t]} p(s)\right) \leqslant e^{-\frac{1}{8}\sum_{s \in [t]} p(s)} \leqslant e^{-\frac{1}{8}C_2 t^\alpha} = e^{-C_4 t^\alpha}$$

### 5.1.5 Putting everything together

Adding up the four previous bounds, for all $t \geqslant C_5, \mathbb{P}(\mathfrak{A}_t) \geqslant 1 - 4dt^{-2} - t^{-1}(\ln t)^{-2} - e^{-C_4 t^\alpha}$.

## 5.2 Thompson sample for the optimal action on clean run

We have already established that clean runs occur with high probability, and now we concentrate on how the algorithm behaves on those runs. Proposition 3 further shows that the optimal action will be selected numerous times when a clean run occurs. In turn, the Thompson sample of the optimal action will be arbitrarily close to its expected reward. This argument is the cornerstone of our analysis (that we believe to be missing in the previous analysis of [20]) and will allow us to control the transient behaviour of the algorithm.

**Proposition 3.** *For* $t \geqslant P_1(m, d, \frac{1}{\Delta_{\min}}, \sigma)$*, if* $\mathfrak{A}_t$ *occurs, then we must have* $M_{A^\star}(t) \geqslant C_6 t^\alpha$ *and* $A^{\star\top}\theta(t) \geqslant A^{\star\top}\mu^\star - h(t)$*. With* $P_1$ *a polynomial in* $m, d, \frac{1}{\Delta_{\min}}, \sigma$*,* $C_6 = C_4/2$*, and* $h(t) = C_1 \sigma m\sqrt{\frac{m \ln t}{C_6 t^\alpha}}$*. It is noted that* $\lim_{t \to \infty} h(t) = 0$

**Proof:** Let us consider a clean run. We can count the number of times the optimal action was not chosen and the variance term of the action is greater than $\Delta_{\min}$:

$$\left|\left\{C_1\sigma\sqrt{m \ln t}A^\top(s)V^{\frac{1}{2}}(s)A(s) \geqslant \Delta_{\min}\right\}\right| \leqslant \sum_{i \in [d]} \left|\left\{i \in A(s), C_1\sigma\sqrt{\frac{m \ln t}{N_i(s)}} \geqslant \frac{\Delta_{\min}}{m}\right\}\right|$$

$$\leqslant d\frac{C_1^2\sigma^2 m^3 \ln t}{\Delta_{\min}^2}$$

And since we have a clean run: $|\{s \in [t] : A^{\star\top}\theta(s) \geqslant A^{\star\top}\mu^\star\}| \geqslant C_2 t^\alpha$. At those times, if the variance term of the action played is less than $\Delta_{\min}$, then it means that the optimal action has been played due to the first condition of $\mathfrak{A}_t$. So we get :

$$M_{A^\star}(t) > C_2 t^\alpha - d\frac{C_1^2\sigma^2 m^3 \ln t}{\Delta_{\min}^2} \geqslant C_6 t^\alpha$$

With $C_6 = C_2/2$ for $t \geqslant P_1(m, d, \frac{1}{\Delta_{\min}}, \sigma) := \left(\frac{1}{\alpha}\right)^{1+\frac{2}{\alpha}} \left(1 - \frac{1}{e}\right)^{-1/\alpha} \left(\frac{C_1^2}{C_2}\right)^{1+1/\alpha} \left(\frac{d\sigma^2 m^3}{\Delta_{\min}^2}\right)^{1+1/\alpha}$ using lemma f. 8. Recall that under $\mathfrak{A}_t$ :

$$|A^{\star\top}\theta(t) - A^{\star\top}\mu^\star| \leqslant C_1\sigma\sqrt{m\ln t}A^\top V^{1/2}(s)A.$$

Since $M_{A^\star}(t) \geqslant C_6 t^\alpha$ has been selected at least $C_6 t^\alpha$ times up to time $t$, we have $A^{\star\top}V^{1/2}A^\star \leqslant \frac{m}{\sqrt{C_6 t^\alpha}}$ so we get the announced result :

$$|A^{\star\top}\theta(t) - A^{\star\top}\mu^\star| \leqslant C_1\sigma m\sqrt{\frac{m\ln t}{C_6 t^\alpha}} = h(t)$$

## 5.3 Regret upper bound

We can now analyze the regret. Let us define some more events at time $t$:

$$\mathfrak{Z}_t := \{\Delta(t) > 0\} \qquad\qquad \mathfrak{H}_t := \left\{A^\top(t)(\theta(t) - \widehat{\mu}(t)) > \sigma\sqrt{8\widetilde{f}(t)A^\top(t)V(t)A(t)}\right\}$$

$$\mathfrak{G}_t := \left\{A^\top(t)(\theta(t) - \mu^\star) > \frac{3\Delta(t)}{4}\right\} \quad \mathfrak{F}_t := \left\{\exists i \in A(t),\, \widehat{\mu}_i(t) - \mu_i^\star > \frac{\Delta_{\min}}{4m}\right\}$$

with $\widetilde{f}(t) := 2\left(\ln(|\mathcal{A}|t) + (m+2)(1 + d\ln 2)\ln(\ln t) + \frac{m(1+d\ln 2)}{2}\ln(1+e)\right)$.

The event $\mathfrak{Z}_t$ means a suboptimal play. $\mathfrak{F}_t$ implies that the empirical mean of one of the items in the action selected at time $t$ deviates from its expectation. $\mathfrak{G}_t$ means that the Thompson sample of the decision played is far from its true value, and finally, $\mathfrak{H}_t$ is for when the Thompson sample from the arm played is far from its empirical mean. The complete event system decomposed as follows.

### 5.3.1 Regret due to $\bar{\mathfrak{A}}_t$

Using proposition 2 we have that $\mathbb{P}(\bar{\mathfrak{A}}_t) \leqslant 4dt^{-2} + t^{-1}(\ln t)^{-2} + e^{-C_4 t^\alpha}$. Then we can use the fact that $\sum_{t\in\mathbb{N}^\star}\frac{1}{t^2} = \frac{\pi^2}{6}, \sum_{t\in\mathbb{N}^\star}\frac{1}{t(\ln t)^2} < 4$. And furthermore, with lemma f. 9 we have that $\sum_{t\in\mathbb{N}^\star}e^{-C_4 t^\alpha} < \frac{C_4^{-1/\alpha}}{\alpha}\Gamma(\frac{1}{\alpha})$. Therefore, the regret caused by $\bar{\mathfrak{A}}_t$ is upper bounded by:

$$\sum_{t\in[T]}\mathbb{E}[\Delta(t)\mathbf{1}\{\bar{\mathfrak{A}}_t\}] < \Delta_{\max}\sum_{t\in[T]}\mathbb{P}(\bar{\mathfrak{A}}_t) < \Delta_{\max}\left[d\frac{2\pi^2}{3} + \frac{C_4^{-1/\alpha}}{\alpha}\Gamma\left(\frac{1}{\alpha}\right) + 4\right].$$

### 5.3.2 Regret due to $\bar{\mathfrak{G}}_t \cap \mathfrak{A}_t$

We use proposition 3 and we get that for $t \geqslant P_1(m, d, 1/\Delta_{\min}, \sigma)$, $A^{\star\top}\theta(t) > A^{\star\top}\mu^\star - h(t)$. Combining with event $\bar{\mathfrak{G}}_t$, we know that when $h(t) < \Delta_{\min}/4$, the only action that can be played is the optimal one. We recall the formula of $h(t) = C_1\sigma m\sqrt{\frac{m\ln t}{C_6 t^\alpha}}$ And using lemma f. 8, this happens for $t > P_2(m, \frac{1}{\Delta_{\min}}, \sigma) := \left(\frac{1}{\alpha}\right)^{1+2/\alpha}\left(1 - \frac{1}{e}\right)^{-1/\alpha}\left(\frac{16C_1^2}{C_6}\right)^{1+1/\alpha}\left(\frac{\sigma^2 m^3}{\Delta_{\min}^2}\right)^{1+1/\alpha}$ So the regret caused by this term is upper bounded by :

$$\sum_{t\in[T]}\mathbb{E}[\Delta(t)\mathbf{1}\{\bar{\mathfrak{G}}_t \cap \mathfrak{A}_t\}] < \Delta_{\max}\max\left\{P_1\left(m, d, 1/\Delta_{\min}, \sigma\right), P_2\left(m, 1/\Delta_{\min}, \sigma\right)\right\}.$$

### 5.3.3 Regret due to $\mathfrak{F}_t$

This result comes from lemma 2 from [6] and is reproduced here in lemma 8. By setting $\epsilon = \frac{\Delta_{\min}}{4}$, we have

$$\sum_{t\in[T]}\mathbb{E}[\Delta(t)\mathbf{1}\{\mathfrak{F}_t\}] < d\Delta_{\max}\left(\frac{32m^2\sigma^2}{\Delta_{\min}^2}\right).$$

### 5.3.4 Regret due to $\mathfrak{H}_t$

We show in lemma 9 in the appendix that $\mathbb{P}(\mathfrak{H}_t) < \frac{1}{t^2}$ and $\sum_{t\in[T]}\mathbb{E}\left[\Delta(t)\mathbf{1}\{\mathfrak{H}_t\}\right] < \Delta_{\max}\frac{\pi^2}{6}$.

### 5.3.5 Regret due to $\bar{\bar{\mathfrak{F}}}_t \cap \mathfrak{G}_t \cap \bar{\mathfrak{H}}_t$

We use lemma 4 in this appendix, and we get with $C = 768, C' = 2304 \ln 2$ that :

$$\sum_{t \in [T]} \mathbb{E}[\Delta(t)\mathbf{1}\left\{\bar{\bar{\mathfrak{F}}}_t \cap \mathfrak{G}_t \cap \bar{\mathfrak{H}}_t\right\}] \leqslant \frac{384\sigma^2 d \ln m \tilde{f}(T)}{\Delta_{\min}} \text{ and }$$

$$\frac{384\sigma^2 d \ln m \tilde{f}(T)}{\Delta_{\min}} < C\frac{\sigma^2 d \ln m}{\Delta_{\min}} \ln T + C'\frac{\sigma^2 d^2 m \ln m}{\Delta_{\min}} \ln \ln T + 1152\frac{\sigma^2 m d^2 \ln 2 \ln(1+e)}{\Delta_{\min}}.$$

### 5.3.6 Putting everything together

Finally, we can put everything together and obtain the regret upper bound found in 1 with the following polynomial constant term :

$$P\left(m, d, \frac{1}{\Delta_{\min}}, \Delta_{\max}, \sigma\right) = \Delta_{\max}\left[\frac{C_4^{-1/\alpha}}{\alpha}\Gamma\left(\frac{1}{\alpha}\right) + 4\right] + d\Delta_{\max}\left(\frac{32m^2\sigma^2}{\Delta_{\min}^2} + \frac{5\pi^2}{3}\right)$$

$$+ \Delta_{\max}\left(P_1\left(m, d, \frac{1}{\Delta_{\min}}, \sigma\right) + P_2\left(m, \frac{1}{\Delta_{\min}}, \sigma\right)\right) + 1152\frac{\sigma^2 m d^2 \ln 2 \ln(1+e)}{\Delta_{\min}}.$$

The degree of this polynomial depends on $1 + 1/\alpha < 10$ with $\alpha = 0.131$. So the degrees of the polynomial in $m, d, 1/\Delta_{\min}, \sigma, \Delta_{\max}$ are respectively $30, 10, 20, 20, 1$.

## 6 Numerical experiments

In this section we perform numerical experiments with Beta CTS, BG-CTS and ESCB on a case where there are only two actions $\mathcal{A} = \left\{A^1, A^2\right\}$ of size $m = d/2$ with $A^1 = (1, ..., 1, 0, ..., 0)$ and $A^2 = (0, ..., 0, 1, ..., 1)$. This action set exhibited exponential regret in [24]. We set $\mu^\star = (0.7, ..., 0.7, 0.9, ..., 0.9)$ with a Bernoulli distribution. The algorithm Beta CTS Uniform prior is initialized with the uniform distribution, while the Beta CTS Jeffreys is initialized with the Jeffreys prior on $[0, 1]$, which puts more weight around the extremities of $[0, 1]$, increasing exploration.

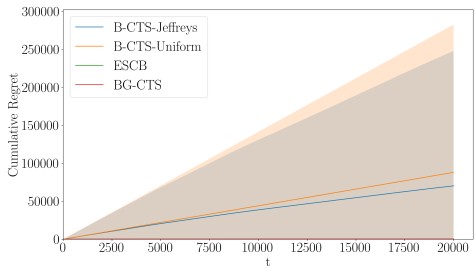

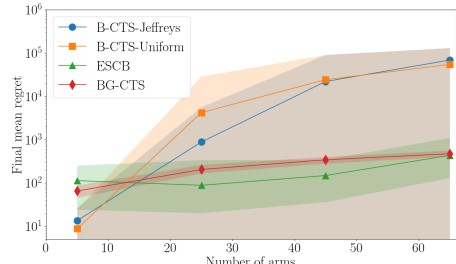

|  (a) Average regret over time | (b) Average final regret as a function of $m$ |

In the first experiment, we set a time horizon of $T = 2 \times 10^4$, and each decision has $m = 50$ items. We run the experiment 100 times and plot the average regret over time and two empirical standard deviations in Figure 1a. The regret is nearly linear for the Beta-based Thompson samplings, whereas the subgaussian Thompson sampling and ESCB showcase regret of magnitude much lower. We set a time horizon in the second experiment $T = 1 \times 10^4$. For each decision size $m \in \{5, 25, 45, 65\}$, we run the experiments 150 times, and we plot the final regret as a function of $m$ in Figure 1b. In the Beta-based Thompson samplings, the final regret and its variance rapidly increase with $m$. In comparison, BG-CTS and ESCB do not seem to be affected.

## 7 Conclusion

We proposed a Boosted variance Gaussian Thompson Sampling for linear combinatorial bandits (BG-CTS) and proved using novel strategies that its regret is bounded polynomially. This variant of TS far outperforms the classical TS by several orders of magnitude on a 2 decisions Bernoulli reward example.

## Acknowledgements

This paper is supported by the CHIST-ERA Wireless AI 2022 call MLDR project (ANR-23-CHR4-0005), partially funded by AEI and NCN under projects PCI2023-145958-2 and 2023/05/Y/ST7/00004, respectively.

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

# A   Glossary

**List notation related to problem setting**

| Notation | Description | Page List |
|---|---|---|
| $[T]$ | Set of time step until $T : \{1, 2, ..., T\}$ | 1 |
| $\mathcal{A}$ | Combinatorial set of interest include in $\{0, 1\}^d$ | 1 |
| $A(t)$ | Action taken at time $t$ | 1 |
| $d$ | Number of arms, dimension of the problem | 1 |
| $m$ | Maximal size of a decision | 1 |
| $X(t)$ | Random vector of dimension $d$ representing the reward of each arm at time $t$ | 1 |
| $Y(t)$ | Observation made by the learner at time $t : Y(t) := A(t) \odot X(t)$ | 1 |
| $\odot$ | The Hadamard operator of two vectors or matrix $A,B$ of the same size. $(A \odot B)_{i,j} := A_{i,j} B_{i,j}$ | 1 |
| $\mu^\star$ | The true unknown parameters | 1 |
| $A^\star$ | The optimal action | 1 |
| $\Delta_A$ | The reward gap between the optimal decision and the decision $A$ | 1 |
| $\Delta_{\min}$ | The minimal reward gap, the reward gap between the optimal decision and one of the second best decision | 1 |
| $\Delta_{\max}$ | The maximal reward gap, the reward gap between the optimal decision and one of the worse decision | 1 |
| $\Delta(t)$ | Reward gap at time $t$ of the action $A(t)$ | 1 |
| $\sigma$ | Sub-gaussian constant of the problem | 1 |
| $R(T, \mu^\star)$ | Regret at time $T$ for instance $\mu^\star$ | 1 |

**List notations related to the algorithm and its analysis**

| Notation | Description | Page List |
|---|---|---|
| $\widehat{\mu}(t)$ | The empirical mean of the parameters at time t | 3 |
| $\theta(t)$ | Sample from the posterior distribution of $\mu^\star$ given by the algorithm at time $t$ | 2 |
| $N(t)$ | Number of times each arm has been selected until time $t$ | 3 |
| $M_A(t)$ | Number of time the action $A$ has been selected until time $t$ | 3 |
| $V(t)$ | Squared diagonal matrix containing the inverse of $N(t)$. The variance of the Thompson samples are proportional to V(t) | 3 |
| $\mathcal{H}(t)$ | History up to time $t$, (The observation at time t is not included) | 4 |
| $\alpha(t)$ | First parameter of the Beta distribution at time $t$ | 4 |
| $\beta(t)$ | Second parameter of the Beta distribution at time $t$ | 4 |
| $g(t)$ | Bonus variance added to the Thompson samples distribution at time $t$ | 4 |
| $f(t)$ | Maximal concentration inequality function at time $t$, given by [10] | 4 |
| $\tilde{f}(t)$ | Upper bound on the quantity $g(t) * \ln(|\mathcal{A}|)$. Controls the deviation random part of the Thompson sample of the arm played. | 8 |
| $h(t)$ | Vanishing term that controls how close the Thompson sample of the optimal action is to its true mean during a clean run | 7 |
| $Z(t)$ | Random i.i.d. Gaussian unitary vector of dimension $d$ folowing $\mathcal{N}(0, I_d)$ used to generate the Thompson samples at time $t$ | 4 |
| $Q$ | Tail function of a standard Gaussian distribution $Q(x) := \mathbb{P}(\mathcal{N}(0, 1) \geqslant x)$ | 6 |
| $U^\star(s)$ | Quantity related to the empirical mean of the best decision at time $s$ | 6 |
| $S^\star(s)$ | Quantity related to the random part of the Thompson sample of the best decision at time $s$ | 6 |
| $W(s)$ | Counting process of the number of times the random part of the Thompson sample of the mean of the best decision deviated | 7 |
| $P_1$ | Polynomial term in $m, d, 1/\Delta_{\min}, \sigma$ that represents the waiting time so that the best action is played more than fractional power of $t$ | 8 |

| Notation | Description | Page List |
|---|---|---|
| $P_2$ | Polynomial term in $m, 1/\Delta_{\min}, \sigma$ that represents the waiting time so that the average mean of the best action is close to its true mean | 8 |
| $P$ | Polynomial term in $m, d, 1/\Delta_{\min}, \sigma$ replacing the exponential term in the previous annalysis of [10] | 9 |

## List of events

| Notation | Description | Page List |
|---|---|---|
| $\mathfrak{A}_t$ | Event of a clean run at time $t$ | 5 |
| $\mathfrak{B}_t$ | The empirical mean deviated too much during the run until time $t$ | 6 |
| $\mathfrak{C}_t$ | The random vector $Z(t)$ deviated to much during the run until time $t$ | 6 |
| $\mathfrak{D}_t$ | The empirical mean of the reward of the optimal action deviated too much during the run until time $t$ | 6 |
| $\mathfrak{E}_t$ | The random part of the Thompson sample of the best decision deviated too few times during the run until time $t$ | 6 |
| $\mathfrak{Z}_t$ | The algorithm plays a suboptimal decision at time $t$ | 8 |
| $\mathfrak{F}_t$ | At time $t$, the empirical mean of one of the arms played is too far from the true mean | 8 |
| $\mathfrak{G}_t$ | At time $t$, the Thompson sample of the decision played is too far from its true mean | 8 |
| $\mathfrak{H}_t$ | At time $t$, the random part of the Thompson sample of the decision played deviated too much | 8 |

## List of the constants used in the paper

| Notation | Description | Page List |
|---|---|---|
| $C_1$ | $C_1 = \sqrt{8} + \sqrt{72}$ | 5 |
| $C_2$ | $C_2 = \frac{1}{2^{13/4 + C_3^2/2} C_3 \sqrt{2\pi}}$ | 5 |
| $C_3$ | $C_3 = \sqrt{1.238}$ | 5 |
| $C_4$ | $C_4 = C_2/8$ | 5 |
| $C_5$ | $C_5 = 23$ | 5 |
| $C_6$ | $C_6 = C_4/2$ | 7 |
| $C$ | $C = 768$ constant in front of the log term in the regret bound | 9 |
| $C'$ | $C' = 2304 \ln(2)$ constant in front of loglog term in the regret bound | 9 |
| $\alpha$ | $\alpha = 3/4 - (1/2)(C_3)^2 \approx 0.131$ | 5 |

## B  Algorithm

Here are the algorithm for Beta Combinatorial Thompson Sampling (Beta CTS) and Boosted Gaussian Combinatorial Thompson Sampling (BG-CTS) that we use in the paper.

---

**Algorithm 1:** Beta Combinatorial Thompson Sampling (Beta CTS) (with a uniform prior)

---

1 **Initialization :** Uniform prior for beta distribution $\alpha(0) = \beta(0) \triangleq [1]^d$ ;
2 **for** $t = 1, ..., T$ **do**
3      Draw $\theta_i(t) \sim \text{Beta}(\alpha_i(t-1), \beta_i(t-1))$
4      Compute $A(t) = \arg\max_{A \in \mathcal{A}}\{A^T\theta(t)\}$
5      The environment draws $X_i(t) \sim \text{Ber}(\mu_i^*)$
6      Observe $X(t) \odot A(t)$, Receive reward $A(t)^T X(t)$
7      Update priors $\alpha(t) = \alpha(t-1) + X(t) \odot A(s)$ and $\beta(t) = \beta(t-1) + (A(t) - X(t) \odot A(t))$
8 **end**

---

**Algorithm 2:** Boosted Gaussian Combinatorial Thompson Sampling (BG-CTS) (with improper prior)

---

**Input:** $\lambda > 0, \sigma > 0$
1 **Initialization :** $\forall i \in [d], N_i(0) = 0, \widehat{\mu}_i(0) = 0$ Uniform improper distribution on $\mathbb{R}^d$, select decisions until $\min_{i \in [d]} N_i(t) > 0$. Update $\forall i \in [d], N_i(0), \mu_i(0)$ accordingly. Generate $\forall t \in [T], \forall i \in [d], Z_i(t) \sim \mathcal{N}(0,1)$ i.i.d.
2 **for** $t = 1, ..., T$ **do**
3      Compute $\theta_i(t) = \widehat{\mu}_i(t-1) + \sigma\sqrt{\frac{2g(t)}{N_i(t-1)}}Z_i(t)$.
4      Compute $A(t) = \arg\max_{A \in \mathcal{A}}\{A^T\theta(t)\}$.
5      The environment draws $\forall i \in [d], X_i(t)$.
6      Observe $X(t) \odot A(t)$, Receive reward $A(t)^T X(t)$.
7      Update $\forall i \in A(t), N_i(t) = N_i(t-1) + 1, \widehat{\mu}_i(t) = \frac{N_i(t)-1}{N_i(t)}\widehat{\mu}_i(t-1) + \frac{X_i(t)}{N_i(t)}$.
8 **end**

---

## C  Proofs of main results

Here in 2 is the diagram of the regret decomposition on the complete event system in green. In red is the novel part of the proof that we introduce. It replaces the step 4 of the proof in [20] wich was inspired by [23] who we think addapted ideas from [2] and [14]. We, in some sense, rediscovered those ideas for the case of combinatorial bandits by controling the numbre of times the optimal action is played

**Lemma 4.** *We have :*

$$\bar{\mathfrak{F}}_t \cap \mathfrak{G}_t \cap \bar{\mathfrak{H}}_t \subset \left\{\Delta_t < 2\sigma\sqrt{8\tilde{f}(t)A^\top(t)V(t)A(t)}\right\}.$$

*And therefore (This is lemma 4 from [19]):*

$$\sum_{t \in [T]} \mathbb{E}[\Delta_t \mathbf{1}\{\bar{\mathfrak{F}}_t \cap \mathfrak{G}_t \cap \bar{\mathfrak{H}}_t\}] \leqslant \sum_{t \in [T]} \mathbb{E}\left[\Delta(t)\mathbf{1}\left\{\Delta_t < 2\sigma\sqrt{8\tilde{f}(t)A^\top(t)V(t)A(t)}\right\}\right].$$

$$\leqslant 384\sigma^2 \ln m\tilde{f}(T) \sum_{i \in [d]} \frac{1}{\Delta_{i,\min}}.$$

*And*

$$\frac{500\sigma^2 d \ln m\tilde{f}(T)}{\Delta_{\min}} < C\frac{\sigma^2 d \ln m}{\Delta_{\min}}\ln T + C'\frac{\sigma^2 d^2 m \ln m}{\Delta_{\min}}\ln\ln T + 1152\frac{md^2 \ln 2\ln(1+e)}{\Delta_{\min}}$$

*with* $C = 768, C' = 2304 \ln 2$.

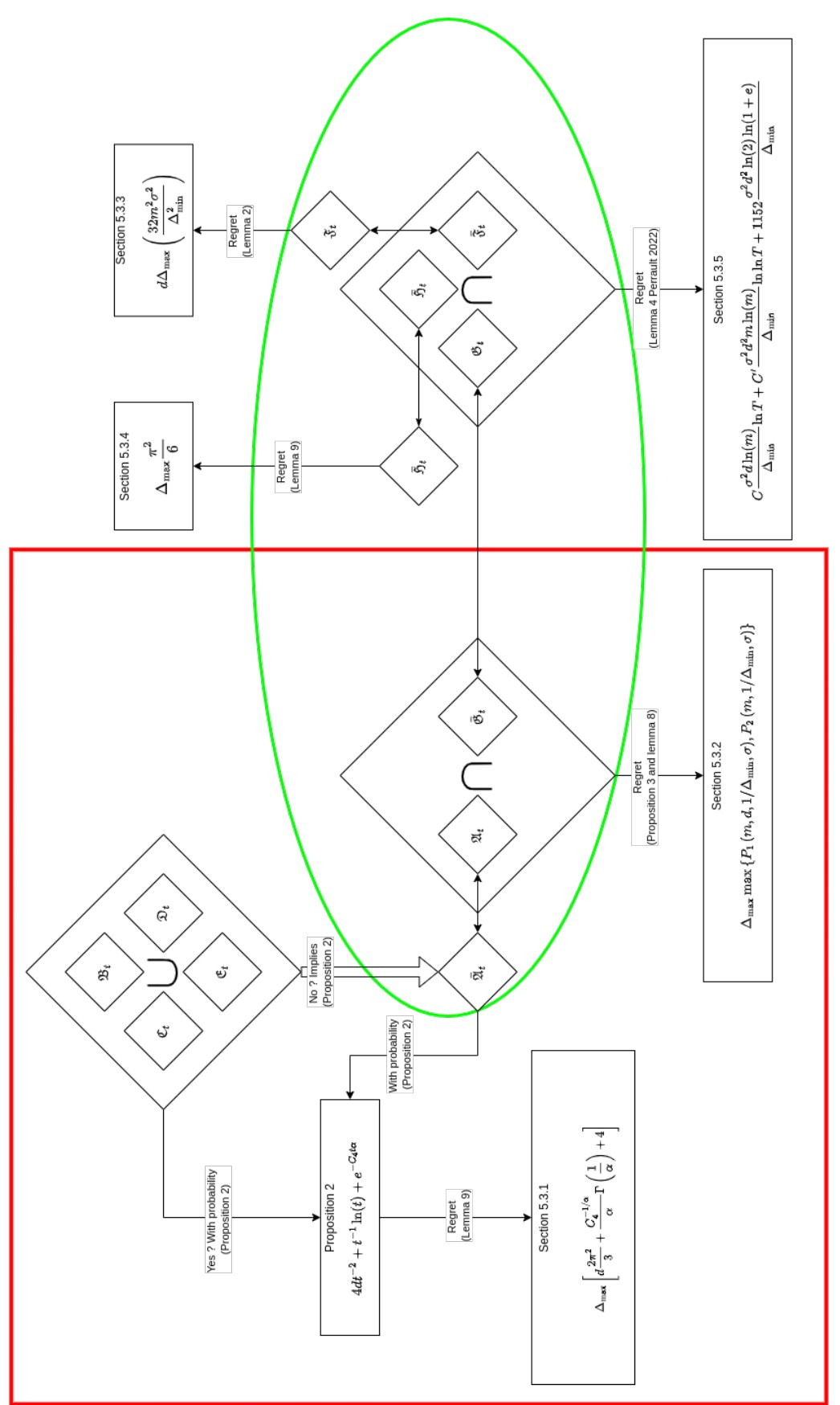

Figure 2: Diagram of the proof of the main result

*Proof.* $\mathfrak{G}_t$ implies that we have

$$\frac{3\Delta(t)}{4} \leqslant \sum_{i \in A(t)} (\theta_i(t) - \mu_i^\star)$$

$$\leqslant \sum_{i \in A(t)} (\theta_i(t) - \widehat{\mu}_i(t)) + \sum_{i \in A(t)} (\widehat{\mu}_i(t-1) - \mu_i^\star)$$

Then $\overline{\overline{\mathfrak{F}}}_t$ respectively $\overline{\mathfrak{H}}_t$ imply that $\sum\limits_{i \in A(t)} (\widehat{\mu}_i(t) - \mu_i^\star) < \frac{\Delta_t}{4}$ respectively $\sum\limits_{i \in A(t)} (\theta_i(t) - \widehat{\mu}_i(t)) < \sigma\sqrt{8\tilde{f}(t)A^\top(t)V(t)A(t)}$. Therefore, we have that :

$$\Delta(t) < 2\sigma\sqrt{8\tilde{f}(t)A^\top(t)V(t)A(t)}$$

Then because $\tilde{f}$ is increasing, we can use lemma 4 from [19] with $\beta_{i,T} = 64\sigma^2\tilde{f}(T)$. And we have that $\forall i, p_i = 1$ because we are not working on triggering bandits. ( $\beta_{i,T}$ and $p_i$ are from their notation, We can also notice that because each arm is played at least once in our algorithm we do not have the first term in $d\Delta_{\max}$ that is counted elsewhere) We have that :

$$\sum_{t \in [T]} \mathbb{E}[\Delta(t)\mathbf{1}\left\{\overline{\overline{\mathfrak{F}}}_t \cap \mathfrak{G}_t \cap \overline{\mathfrak{H}}_t\right\}]$$

$$\leqslant \sum_{t \in [T]} \mathbb{E}\left[\Delta(t)\mathbf{1}\left\{\Delta(t) < 2\sigma\sqrt{8\tilde{f}(T)A^\top(t)V(t)A(t)}\right\}\right]$$

$$\leqslant 64\sigma^2\tilde{f}(T)\sum_{i \in [d]}\frac{3 + \ln(m)}{\Delta_{i,\min}}.$$

To make the formula more readable we use that for $m > 2$ we have that $5\ln m > 3$ so that :

$$64\sigma^2\tilde{f}(T)\sum_{i \in [d]}\frac{3 + \ln(m)}{\Delta_{i,\min}} < 384\sigma^2\tilde{f}(T)\sum_{i \in [d]}\frac{\ln(m)}{\Delta_{i,\min}}$$

By definition of $\tilde{f}$, we have that $C = 768$. And assuming $m \geqslant 1, d \geqslant 3$ we have that $(m+2)(1 + d\ln 2) < 3dm\ln 2$ so $C'$ can be taken as $2304\ln 2$ and the last constant is $1152\ln(2)$.

$\square$

## D  Concentration results

**Lemma 5.** *For $t \geqslant 2$, let $\lambda > 0$, let $\delta_t > 0$. Let $f(\tilde{\delta}_t) := \ln(\frac{1}{\delta_t}) + m\ln\ln t + \frac{m}{2}\ln\left(1 + \frac{e}{\lambda}\right)$. Then the event $\mathfrak{D}_t = \{\max_{s \in [t]} U^\star(s) \geqslant \sigma\sqrt{2(1 + \lambda)f(\delta_t)}\}$ happen with probability $\mathbb{P}(\mathfrak{D}_t) < \delta_t$*

*Proof.* See lemma 3 of [10]. Moreover, notice that in their lemma 6, they use a union bound on all the possible values of $(N_i)_{i \in A^\star} \in [t]^m$ to get their result. (This is symbolized by their set $\mathcal{D}_a$) $\square$

**Lemma 6** (Multiplicative Azuma Chernoff). *Let $(W_t)_{t \in \mathbb{N}^\star}$ be a sum of random variable : $\forall t \in \mathbb{N}^\star, W_t = \sum\limits_{s=1}^{t} X_s$ Where $(X_t)_{t \in N^\star}$ verify that there exist $(p_t)_{t \in N} \in ]0,1[$ such that :*

$$\forall t \in N^\star, \mathbb{P}(X_t = 1|\mathcal{H}(t-1)) = p_t,$$
$$\mathbb{P}(X_t = 0|\mathcal{H}(t-1)) = 1 - p_t.$$

*And $(\mathcal{H}(t))_{t \in \mathbb{N}^\star}$ is a filtration where $\forall t \in \mathbb{N}^\star, X_t$ is $\mathcal{H}(t)$ measurable. We note $m_t = \sum\limits_{t=1}^{t} p_{t'}$.*

*Let $t \in \mathbb{N}^\star, \forall \delta > 0$ we have that :*

$$\mathbb{P}\left(W_t \geqslant (1 + \delta)m_t\right) < \exp(\frac{\delta^2 m_t}{2 + \delta})$$

*and $\forall\, 1 > \delta \geqslant 0$*

$$\mathbb{P}\left(W_t \leqslant (1 - \delta)m_t\right) < \exp(\frac{\delta^2 m_t}{2})$$

*Proof.* **For the first inequality**, let $t \in \mathbb{N}^\star, \lambda, x \in \mathbb{R}^+$, we have by Markov inequality:

$$\begin{aligned}
\mathbb{P}(W_t \geqslant x) &= \mathbb{P}(\lambda W_t \geqslant \lambda x) \\
&= \mathbb{P}(e^{\lambda W_t} \geqslant e^{\lambda x}) \\
&\leqslant \mathbb{E}(e^{\lambda W_t})e^{-\lambda x}
\end{aligned}$$

Let $s \in [t]$, by definition of $X_s$ and using that $\forall x \in \mathbb{R}, \exp(x) \geqslant 1 + x$ we have that :

$$\begin{aligned}
\mathbb{E}[e^{\lambda X_s} | \mathcal{H}(s - 1)] &= p_s e^\lambda + (1 - p_s) \\
&= p_s(e^\lambda - 1) + 1 \\
&\leqslant \exp(p_s(e^\lambda - 1)).
\end{aligned}$$

Then, by the tower property of the expectation and induction:

$$\begin{aligned}
\mathbb{E}(e^{\lambda W_t}) &= \mathbb{E}\left[e^{\lambda W_{t-1}}\mathbb{E}\left[e^{\lambda X_t}|\mathcal{H}_{t-1}\right]\right] \\
&\leqslant \mathbb{E}\left[e^{\lambda W_{t-1}}\right]\exp(p_s(e^\lambda - 1)) \\
&\leqslant \exp\left(\sum_{s \in [t]} p_s(e^\lambda - 1)\right).
\end{aligned}$$

Combining with the first equation, we have :

$$\mathbb{P}(W_t > x) < \exp\left(m_t(e^\lambda - 1) - \lambda x\right)$$

This is true for all $x, \lambda \in \mathbb{R}^+$ so by setting $\lambda = \ln(1 + \delta)$ and $x = (1 + \delta)m_t$ we have :

$$\begin{aligned}
\mathbb{P}(W_t > (1 + \delta)m_t) &< \exp\left(m_t(e^{\ln(1+\delta)} - 1) - (1 + \delta)\ln(1 + \delta)m_t\right) \\
&\leqslant \exp(m_t(\delta - (1 + \delta)\ln(1 + \delta))) \\
&\leqslant \exp(-\frac{\delta^2 m_t}{2 + \delta})
\end{aligned}$$

Because $\forall \delta > 0, \delta - (1 + \delta)\ln(1 + \delta) < \frac{-\delta^2}{2+\delta}$.

**For the second inequality**, let $\lambda \in \mathbb{R}^+$:

$$\begin{aligned}
\mathbb{P}(W_t < x) &= \mathbb{P}(-\lambda W_t > -\lambda x) \\
&= \mathbb{P}(e^{-\lambda W_t} > e^{-\lambda x}) \\
&\leqslant \mathbb{E}(e^{-\lambda W_t})e^{\lambda x}
\end{aligned}$$

Let $s \in [t]$, by definition of $X_s$ we have that :

$$\begin{aligned}
\mathbb{E}[e^{-\lambda X_s} | \mathcal{H}(s - 1)] &= p_s e^{-\lambda} + (1 - p_s) \\
&= p_s(e^{-\lambda} - 1) + 1 \\
&\leqslant \exp(p_s(e^{-\lambda} - 1))
\end{aligned}$$

Then, by the tower property of the expectation and induction:

$$\mathbb{E}(e^{-\lambda W_t}) = \mathbb{E}\left[e^{-\lambda W_{t-1}}\mathbb{E}\left[e^{-\lambda X_t}|\mathcal{H}_{t-1}\right]\right]$$

$$\leqslant \mathbb{E}\left[e^{-\lambda W_{t-1}}\right]\exp(p_s(e^{-\lambda}-1))$$

$$\leqslant \exp\left(\sum_{s\in[t]}p_s(e^{-\lambda}-1)\right)$$

Combining with the first equation, we have :

$$\mathbb{P}(W_t < x) < \exp\left(m_t(e^{-\lambda}-1) + \lambda x\right)$$

This is true for all $x, \lambda \in \mathbb{R}^+$ so by setting $\lambda = -\ln(1-\delta)$ and $x = (1-\delta)m_t$ we have :

$$\mathbb{P}(W_t < (1-\delta)m_t) < \exp\left(m_t(e^{\ln(1-\delta)}-1) - (1-\delta)\ln(1-\delta)m_t\right)$$

$$\leqslant \exp(m_t(-\delta - (1-\delta)\ln(1-\delta)))$$

$$\leqslant \left(\frac{e^{-\delta}}{(1-\delta)^{(1-\delta)}}\right)^{m_t}$$

$$\leqslant e^{-\frac{\delta^2 m_t}{2}}$$

$\square$

**Lemma 7.** *Let $t \in [T]$, let us consider the event :*

$$\mathfrak{B}(t) := \left\{\max_{s\in[t]}\|V^{\frac{1}{2}}(s)(\mu^\star - \widehat{\mu}(s))\|_\infty > \sigma\sqrt{8\ln t}\right\}.$$

*We have that $\mathbb{P}(\mathfrak{B}(t)) < \frac{d}{t^2}$. This result can be in part found in [16] in their proof of lemma 1.*

*Proof.* We control it with the deviation of individual arms.

$$\mathfrak{B}_i(t) := \left\{\exists s \in [t], |\mu_i^\star - \widehat{\mu}_i(s)| > \sigma\sqrt{\frac{8\ln t}{N_i(s)}}\right\}$$

We have by a double union bound, and because the rewards are $\sigma^2$ subgaussian using Hoeffding:

$$\mathbb{P}(\mathfrak{B}_i(t)) = \sum_{s\in[t]}\sum_{n\in[s]}\mathbb{P}(|\mu_i^\star - \widehat{\mu}_i(s)| > \sigma\sqrt{\frac{8\ln t}{n}}, N_i(s) = n)$$

$$< \sum_{s\in[t]}\sum_{n\in[s]}\mathbb{P}(|\sum_{k\in[n]}(X_i(k) - \mu_i^\star)| > \sigma\sqrt{8n\ln t})$$

$$< 2t^2\exp(-4\ln t)$$

$$< \frac{2}{t^2}$$

We have that $\mathfrak{B}(t) \subset \bigcup_{i\in d}\mathcal{M}_i(t)$. So by union bound $\mathbb{P}(\mathfrak{B}(t)) < \frac{2d}{t^2}$. $\square$

**Lemma 8.** *Let $i \in [d], t \in [T]$, let us consider the event :*

$$\mathfrak{F}(t) := \left\{\exists i \in A(t), \; \widehat{\mu}_i(t) - \mu_i^\star > \frac{\Delta_{\min}}{2m} - \frac{\epsilon}{m}\right\}$$

*We have :*

$$\mathbb{E}\left[\sum_{t\in[T]}\mathbf{1}\left\{i \in A(t), \widehat{\mu}_i(t) - \mu_i^\star > \frac{\Delta_{\min}}{2m} - \frac{\epsilon}{m}\right\}\right] < \frac{8m^2\sigma^2}{(\Delta_{\min} - 2\epsilon)^2}$$

*And :*

$$\mathbb{E}\left[\sum_{t\in[T]}\Delta(t)\mathbf{1}\left\{\mathfrak{F}(t)\right\}\right] < d\Delta_{\max}\left(\frac{8m^2\sigma^2}{(\Delta_{\min} - 2\epsilon)^2}\right)$$

*This is a result from [6] lemma 2 adapted to the $\sigma^2$ subgaussian case.*

*Proof.* We use a union bound and Hoeffding's inequality for $\sigma^2$ subgaussian random variable. Let $i \in [d]$,

$$
\mathbb{E}\left[ \sum_{t \in [T]} \mathbf{1}\left\{ i \in A(t), \widehat{\mu}_i(t) - \mu_i^\star > \frac{\Delta_{\min}}{2m} - \frac{\epsilon}{m} \right\} \right]
$$

$$
< \mathbb{E}\left[ \sum_{t=1}^{\infty} \mathbf{1}\left\{ i \in A(t), \widehat{\mu}_i(t) - \mu_i^\star > \frac{\Delta_{\min}}{2m} - \frac{\epsilon}{m} \right\} \right]
$$

$$
< \mathbb{E}\left[ \sum_{n=1}^{\infty} \mathbf{1}\left\{ \widehat{\mu}_i - \mu_i^\star > \frac{\Delta_{\min}}{2m} - \frac{\epsilon}{m}, N_i = n \right\} \right]
$$

$$
< \sum_{n=1}^{\infty} \mathbb{P}\left( \widehat{\mu}_i - \mu_i^\star > \frac{\Delta_{\min}}{2m} - \frac{\epsilon}{m}, N_i = n \right)
$$

$$
< \sum_{n=1}^{\infty} \exp\left( -\frac{n}{2\sigma^2} \left( \frac{\Delta_{\min}}{2m} - \frac{\epsilon}{m} \right)^2 \right)
$$

$$
< \frac{\exp\left( -\frac{1}{2\sigma^2} \left( \frac{\Delta_{\min}}{2m} - \frac{\epsilon}{m} \right)^2 \right)}{1 - \exp\left( -\frac{1}{2\sigma^2} \left( \frac{\Delta_{\min}}{2m} - \frac{\epsilon}{m} \right)^2 \right)}
$$

$$
< \frac{8m^2\sigma^2}{(\Delta_{\min} - 2\epsilon)^2}.
$$

Then decomposing $\mathfrak{F}(t)$ with a union bound of the $\left\{ i \in A(t), \widehat{\mu}_i(t) - \mu_i^\star > \frac{\Delta_{\min}}{2m} - \frac{\epsilon}{m} \right\}$ and summing over $i \in [d]$ we get the second result.

$\square$

**Lemma 9.** *Let $t \in [T]$ we have :*

$$
\mathbb{P}\left( A^\top(t)(\theta(t) - \widehat{\mu}(t)) \geqslant \sigma\sqrt{8g(t)\ln\left(|\mathcal{A}|t\right) A^\top(t)V(t)A(t)} | \mathcal{H}(t) \right) < \frac{1}{t^2}
$$

*Which implies that $\mathbb{P}(\mathfrak{H}(t)|\mathcal{H}(t)) < \frac{1}{t^2}$.*
*Defining :*

$$
\tilde{f}(t) := (1 + \lambda)\left( \ln\left(|\mathcal{A}|t\right) + (m+2)(1 + d\ln(2))\ln(\ln t) + \frac{m(1+d\ln(2))}{2}\ln\left(1 + \frac{e}{\lambda}\right) \right),
$$

*we have that $g(t)\ln(|\mathcal{A}|t) < \tilde{f}(t)$ and thus:*

$$
\mathbb{P}\left( A^\top(t)(\theta(t) - \widehat{\mu}(t)) \geqslant \sigma\sqrt{8\tilde{f}(t)A^\top(t)V(t)A(t)} | \mathcal{H}(t) \right) < \frac{1}{t^2}
$$

*Finally:*

$$
\mathbb{E}\left[ \sum_{t \in [T]} \Delta(t)\mathbf{1}\left\{ \mathfrak{H}(t) \right\} \right] < \Delta_{\max} \frac{\pi^2}{6}
$$

*Proof.* Let $c > 0$ by union bound we have :

$$\mathbb{P}\left(A^\top(t)(\theta(t) - \widehat{\mu}(t)) \geqslant c\sigma\sqrt{2g(t)A^\top(t)V(t)A(t)}|\mathcal{H}(t)\right)$$

$$< \sum_{A \in \mathcal{A}} \mathbb{P}\left(\frac{A^\top(t)(\theta(t) - \widehat{\mu}(t))}{\sigma\sqrt{2g(t)A^\top(t)V(t)A(t)}} \geqslant c|\mathcal{H}(t)\right)$$

$$< \sum_{A \in \mathcal{A}} \mathbb{P}\left(\frac{\sigma\sqrt{2g(t)}A^\top(t)V^{\frac{1}{2}}(t)Z(t)}{\sigma\sqrt{2g(t)A^\top(t)V(t)A(t)}} \geqslant c|\mathcal{H}(t)\right)$$

$$= |\mathcal{A}|Q(c)$$

$$< \frac{|\mathcal{A}|}{c\sqrt{2\pi}}\exp(-\frac{c^2}{2})$$

$$< |\mathcal{A}|\exp(-\frac{c^2}{2})$$

By setting $c = \sqrt{4\ln(|\mathcal{A}|t)} > 1$ we have $|\mathcal{A}|\exp\left(-\frac{c^2}{2}\right) = \frac{1}{t^2}$ hence the first result.

Then we have that : $\ln(|\mathcal{A}|) \leqslant d\ln(2)$ so that :

$$\frac{\ln(|\mathcal{A}|t)}{\ln t} \leqslant \ln(|\mathcal{A}|) + 1$$
$$\leqslant 1 + d\ln(2)$$

Hence, the second result.

And by summing over $t$:

$$\mathbb{E}\left[\sum_{t \in [T]} \Delta(t)\mathbf{1}\{\mathfrak{H}(t)\}\right] < \sum_{t \in [T]} \Delta_{\max}\mathbb{E}\left[\mathbb{E}\left[\mathbf{1}\{\mathfrak{H}(t)\}|\mathcal{H}(t)\right]\right]$$

$$< \sum_{t \in [T]} \Delta_{\max}\mathbb{E}\left[\mathbb{P}(\mathfrak{H}(t))|\mathcal{H}(t)\right]$$

$$< \sum_{t \in [T]} \Delta_{\max}\frac{1}{t^2}$$

$$< \Delta_{\max}\frac{\pi^2}{6}$$

$\square$

We recall simple tail bounds for Gaussian random variables. (see for instance [12])

**Lemma 10.** *Consider $Z \sim N(0,1)$, then we have*

$$\mathbb{P}(Z \geqslant x) = Q(x) = \frac{1}{\sqrt{2\pi}}\int_x^{+\infty} e^{-\frac{z^2}{2}}dz$$

*furthermore for all $x \geqslant 0$:*

$$\frac{1}{\sqrt{2\pi}}\frac{x}{1+x^2}e^{-\frac{z^2}{2}} \leqslant Q(x) \leqslant \frac{1}{\sqrt{2\pi}}\frac{1}{x}e^{-\frac{x^2}{2}}$$

*As well as:*

$$Q(x) \leqslant e^{-\frac{x^2}{2}}$$

*It is also noted that by symmetry $\mathbb{P}(|Z| \geqslant x) = 2Q(x)$.*

# E  Functions Study

**Lemma f. 1.** *Let $f$ be the following function :*

$$f(t, m, \lambda) := (1 + \lambda) \left( \ln t + (m + 2) \ln \ln t + \frac{m}{2} \ln \left( 1 + \frac{e}{\lambda} \right) \right)$$

*We have*

$$\forall t > 23, m \in \mathbb{N}^\star, \lambda \in \mathbb{R}^+, \frac{f(t, m, \lambda)}{f(\frac{t}{2}, m, \lambda)} \leqslant \frac{f(23, 1, \lambda)}{f(\frac{23}{2}, 1, \lambda)} \leqslant 1.282.$$

*Proof.*

$$\frac{f(t, m, \lambda)}{f(\frac{t}{2}, m, \lambda)} = \frac{(1 + \lambda) \left( \ln t + (m + 2) \ln \ln t + \frac{m}{2} \ln \left( 1 + \frac{e}{\lambda} \right) \right)}{(1 + \lambda) \left( \ln(\frac{t}{2}) + (m + 2) \ln \ln(\frac{t}{2}) + \frac{m}{2} \ln \left( 1 + \frac{e}{\lambda} \right) \right)}$$

Using f. 2 we have that :

$$\frac{f(t, m, \lambda)}{f(\frac{t}{2}, m, \lambda)} \leqslant \frac{\ln t + (m + 2) \ln \ln t}{\ln(\frac{t}{2}) + (m + 2) \ln \ln(\frac{t}{2})}$$

Using f. 3 we have that for $t > 23, \forall m \geqslant 1$

$$\frac{f(t, m, \lambda)}{f(\frac{t}{2}, m, \lambda)} \leqslant \frac{\ln t + 3 \ln \ln t}{\ln(\frac{t}{2}) + 3 \ln \ln(\frac{t}{2})}$$

But thanks to f. 4, the right-hand side is decreasing in $t$, so :

$$\forall t > 23, m \in \mathbb{N}^\star, \lambda \in \mathbb{R}^+, \frac{f(t, m, \lambda)}{f(\frac{t}{2}, m, \lambda)} \leqslant \frac{f(23, 1, \lambda)}{f(\frac{23}{2}, 1, \lambda)} \leqslant 1.282.$$

$\square$

**Lemma f. 2.** *Let $a \geqslant b \geqslant 0$ and let $g : \mathbb{R}^+ \to \mathbb{R}^+$ be the following function :*

$$g(t) := \frac{a + t}{b + t}$$

*Then $\forall t \geqslant 0, g(t) < g(0) < \frac{a}{b}$*

*Proof.* Let $t \in \mathbb{R}^+$, we have :

$$\begin{aligned} g(t) &= \frac{a + t}{b + t} \\ &= \frac{a + b + t}{b + t} - \frac{b}{b + t} \\ &= 1 + \frac{a - b}{b + t} \end{aligned}$$

So $g$ is decreasing in $t$.

$\square$

**Lemma f. 3.** *The function*

$$f(t, m) := \frac{\ln t + (m + 2) \ln \ln t}{\ln(\frac{t}{2}) + (m + 2) \ln \ln(\frac{t}{2})}$$

*is decreasing in $m$ for $t > 23$.*

*Which means that for $t > 23$, for $\forall m \geqslant 1$ :*

$$\frac{\ln t + (m + 2) \ln \ln t}{\ln(\frac{t}{2}) + (m + 2) \ln \ln(\frac{t}{2})} \leqslant \frac{\ln t + \ln \ln t}{\ln(\frac{t}{2}) + 3 \ln \ln(\frac{t}{2})}$$

*Proof.* Let $l(m) = \frac{a+mb}{c+md}$, its derivative is :

$$l'(m) = \frac{b(c+md) - (a+mb)d}{(c+md)^2} = \frac{bc - ad}{(c+md)^2}$$

So $l'(m) > 0$ i.i.f. $bc - ad > 0$. Furthermore, $\ln\ln t \ln(\frac{t}{2}) - \ln\ln(\frac{t}{2})\ln t < 0$ for $t > 23$ So $\forall t > 23$, $f(t, m)$ is decreasing in $m$ and $\forall t > 23$, $f(t, m) < f(t, 1)$.  □

**Lemma f. 4.** *The function :*

$$f(t) := \frac{\ln t + 3\ln\ln t}{\ln(\frac{t}{2}) + 3\ln\ln(\frac{t}{2})},$$

*is decreasing.*

*Proof.* We differentiate, and we obtain

$$f'(t) < 0$$
$$\iff (\frac{1}{t} + \frac{3}{t\ln t})\left(\ln\left(\frac{t}{2}\right) + 3\ln\ln(\frac{t}{2})\right) - (\frac{1}{t} + \frac{3}{t\ln(\frac{t}{2})})(\ln t + 3\ln\ln t) < 0.$$

We expend and simplify :

$$(\frac{1}{t} + \frac{3}{t\ln t})\left(\ln\left(\frac{t}{2}\right) + 3\ln\ln(\frac{t}{2})\right) - (\frac{1}{t} + \frac{3}{t\ln(\frac{t}{2})})(\ln t + 3\ln\ln t)$$
$$= -\frac{\ln(2)}{t} + \left(\frac{3\ln(\frac{t}{2})}{t\ln t} - \frac{3\ln t}{t\ln(\frac{t}{2})}\right)$$
$$+ \left(\frac{3\ln\ln(\frac{t}{2})}{t} - \frac{3\ln\ln t}{t}\right)$$
$$+ \left(\frac{9\ln\ln(\frac{t}{2})}{t\ln t} - \frac{9\ln\ln t}{t\ln(\frac{t}{2})}\right).$$

We have :

$$\frac{3\ln t}{t\ln(\frac{t}{2})} > \frac{3\ln t}{t\ln t} > \frac{3\ln(\frac{t}{2})}{t\ln t}$$

and

$$\frac{3\ln\ln t}{t} > \frac{3\ln\ln(\frac{t}{2})}{t}$$

and

$$\frac{9\ln\ln t}{t\ln(\frac{t}{2})} > \frac{9\ln\ln t}{t\ln t} > \frac{9\ln\ln(\frac{t}{2})}{t\ln t}$$

Therefore $\forall t > 3$, $f'(t) < 0$ and the function $f$ is decreasing.  □

**Lemma f. 5.** *Let $c > 1$ be a positive constant and let $f$ be the following function :*

$$f(t) = Q(c\sqrt{\ln(t/2)}) \geqslant \frac{1}{\sqrt{2\pi}}\frac{c\sqrt{\ln(t/2)}}{1 + c^2\ln(t/2)}(t/2)^{-\frac{c^2}{2}}$$

*Then for $t > 2e$ we have*

$$f(t) \geqslant \frac{1}{\sqrt{2\pi}} \frac{1}{2c} (t/2)^{-\frac{c^2 + \frac{1}{2}}{2}} = \frac{1}{\sqrt{2\pi}} \frac{1}{2^{\frac{5}{4}} 2^{\frac{c^2}{2}} c} t^{-\frac{c^2 + \frac{1}{2}}{2}}$$

*Proof.* For $t > 2e$ we have that $1 < c^2 \ln(t/2)$ so :

$$\frac{c\sqrt{\ln(t/2)}}{1 + c^2 \ln(t/2)} > \frac{1}{2c} \frac{1}{\sqrt{\ln(t/2)}}$$

For $t > 2e$ we have that $\frac{1}{\sqrt{\ln(t/2)}} = \exp\left(-\frac{1}{2} \ln\ln(t/2)\right) > \exp\left(-\frac{1}{4}\ln(t/2)\right)$. Therefore :

$$f(t) \geqslant \frac{1}{\sqrt{2\pi}} \frac{c\sqrt{\ln(t/2)}}{1 + c^2 \ln(t/2)} (t/2)^{-\frac{c^2}{2}} > \frac{1}{\sqrt{2\pi}} \frac{1}{2c} (t/2)^{-\frac{c^2 + \frac{1}{2}}{2}}$$

$\square$

**Lemma f. 6.**

$$\forall t > e, \frac{\ln\ln t}{\ln t} < \frac{1}{2}$$

**Lemma f. 7.** *Let $f, g$ the following function :*

$$f(t, m, \lambda) := (1 + \lambda)\left(\ln t + (m + 2)\ln\ln t + \frac{m}{2}\ln\left(1 + \frac{e}{\lambda}\right)\right)$$

$$g(t) = \frac{f(t)}{\ln t}$$

*For $t > e$ and $t > 1 + \frac{e}{\lambda}$ we have $\frac{\ln\ln t}{\ln t} < \frac{1}{2}$ thus*

$$f(t) < (1 + \lambda)(2m + 1)\ln t,$$

*and*

$$g(t) < (1 + \lambda)(2m + 1).$$

*Proof.* For $t > e, \ln t > \ln\ln t$ and for $t > 1 + \frac{e}{\lambda}, \ln t > \ln(1 + \frac{e}{\lambda})$ thus the result $\square$

**Lemma f. 8.** *Let $\alpha, c \in \mathbb{R}_+^\star$, two strictly positive constants such that $\frac{c}{\alpha} > 1$ and define the function $f(t) := t^\alpha - c\ln t$. We have that $\forall t > \left(\frac{\frac{c}{\alpha}\ln\frac{c}{\alpha}}{1 - \frac{1}{e}}\right)^{\frac{1}{\alpha}}, f(t) > 0$. Furthermore, we have :*
$\left(\frac{\frac{c}{\alpha}\ln\frac{c}{\alpha}}{1 - \frac{1}{e}}\right)^{\frac{1}{\alpha}} < \left(\frac{1}{\alpha}\right)^{1 + \frac{2}{\alpha}}\left(1 - \frac{1}{e}\right)^{-\frac{1}{\alpha}} c^{1 + \frac{1}{\alpha}}$

*Proof.* Let's study the function $G(T) = T - c\ln\left(T^{\frac{1}{\alpha}}\right) = T - \frac{c}{\alpha}\ln(T)$. For $T \geqslant \frac{c}{\alpha}e$ by using the concavity of the logarithm and differentiating $T \mapsto \frac{c}{\alpha}\ln(T)$ at the point $\frac{c}{\alpha}e$ we have that :

$$\frac{c}{\alpha}\ln(T) < \frac{c}{\alpha}\ln\left(\frac{c}{\alpha}e\right) + \left(T - \frac{c}{\alpha}e\right)\frac{1}{e}$$
$$< \frac{T}{e} + \frac{c}{\alpha}\ln\left(\frac{c}{\alpha}\right)$$

Then for $T \geqslant \frac{c}{\alpha}e$

$$T - \frac{c}{\alpha}\ln(T) > T - \frac{T}{e} - \frac{c}{\alpha}\ln\left(\frac{c}{\alpha}\right)$$
$$> \left(1 - \frac{1}{e}\right)T - \frac{c}{\alpha}\ln\left(\frac{c}{\alpha}\right)$$

So for $T > \frac{\frac{c}{\alpha}\ln(\frac{c}{\alpha})}{1-\frac{1}{e}}, T - \frac{c}{\alpha}\ln(T) > 0$. Which means that by a change of variable that for $t^\alpha > \frac{\frac{c}{\alpha}\ln(\frac{c}{\alpha})}{1-\frac{1}{e}} \iff t > (\frac{\frac{c}{\alpha}\ln\frac{c}{\alpha}}{1-\frac{1}{e}})^{\frac{1}{\alpha}}$ we have that $f(t) > 0$. Hence, the first result.

We have using that $\ln(x) < x$ :

$$\left(\ln\frac{c}{\alpha}\right)^{\frac{1}{\alpha}} = \left(\frac{1}{\alpha}\ln\left(\left(\frac{c}{\alpha}\right)^\alpha\right)\right)^{\frac{1}{\alpha}}$$

$$< \left(\frac{1}{\alpha}\right)^{\frac{1}{\alpha}}\frac{c}{\alpha}$$

Finally, combined with the rest, we get the second result.

$\square$

**Lemma f. 9.** *Let $c > 0, \alpha > 0$ the series $\sum_{t=1}^{\mathsf{T}}\exp(-ct^\alpha)$ converges when $T \to +\infty$ and :*

$$\sum_{t=1}^{\infty}\exp(-ct^\alpha) < \frac{c^{-\frac{1}{\alpha}}}{\alpha}\Gamma(\frac{1}{\alpha})$$

*Proof.* Because the function $t \mapsto \exp(-ct^\alpha)$ is decreasing, by an integral test for convergence, we have that :

$$\sum_{t=1}^{\infty}\exp(-ct^\alpha) < \int_{0}^{+\infty}\exp(-ct^\alpha)dt.$$

A primitive of the function : $t \mapsto \exp(-ct^\alpha)$ is $T \mapsto \frac{c^{-\frac{1}{\alpha}}}{\alpha}\int_{cT^\alpha}^{+\infty}t^{\frac{1}{\alpha}-1}e^{-t}dt$. So :

$$\sum_{t=1}^{\infty}\exp(-ct^\alpha) < \frac{c^{-\frac{1}{\alpha}}}{\alpha}\int_{0}^{+\infty}t^{\frac{1}{\alpha}-1}e^{-t}dt - 0$$

$$\leqslant \frac{c^{-\frac{1}{\alpha}}}{\alpha}\Gamma(\frac{1}{\alpha})$$

$\square$

