# OpenReview forum: "Thompson Sampling For Combinatorial Bandits: Polynomial Regret and Mismatched Sampling Paradox"
_NeurIPS.cc/2024/Conference — NeurIPS 2024 spotlight_

### Official Review · Reviewer_yJpu · 2024-06-19

**Soundness:** 3
**Presentation:** 3
**Contribution:** 3
**Rating:** 7
**Confidence:** 5

**Summary:**

The paper explores CTS for the linear combinatorial semi-bandit problem with subgaussian rewards. It introduces a novel TS algorithm that avoids exponential regret scaling with problem dimensionality. Theoretical bounds and experiments are provided.

**Strengths:**

The paper addresses a significant limitation in existing CTS algorithms by avoiding exponential regret scaling with problem dimensionality, which is a notable contribution to the field. It establishes new regret bounds that improve upon previous results in terms of finite-time performance. This is further shown experimentally.

The paper is well-organized and clearly presents its ideas.

**Weaknesses:**

The paper places significant emphasis on the "mismatched sampling paradox," showcasing situations where a divergence between assumed and actual reward distributions can unexpectedly improve performance. While intriguing, this phenomenon isn't entirely novel, as it's known that Thompson Sampling (TS) can adapt to various posterior distributions. The paper's focus on this paradox, including its prominence in the title, may overshadow what could be considered the core contribution: a little boost on the exploration of CTS to better control the regret constant term.

The paper does not sufficiently clarify how the exploration boost is integrated into the analysis. The proof strategy is not well synthesized.

[minor] Your main term scale with log(m)^2, but it is now possible to get a log(m) term instead. For example, use Lemma 4 in https://arxiv.org/pdf/2302.11182. I think this could be good to at least cite this paper mentioning that this could be done. line 203-204, I think the event D_t should contains the norm infty.

**Questions:**

Can you explain better what makes your analysis work exactly ? What are the proof techniques that you used, and how this differs from [18] ? I put a good score for this paper as I think it worth it. But I can consider putting a lower score if a not satisfying answer is given to the questions.

**Limitations:**

The analysis only work for linear reward. Can you discuss why this is the case and how would it be possible to extend it the more general cases?

---

> ### Author Rebuttal · Authors · 2024-08-06
>
> The Lemma 4 of  (https://arxiv.org/pdf/2302.11182) is very interesting. It could replace lemma 10 from [9] and provide us with a better bound with a $\ln(m)$ instead of a $\ln(m)^2$. We will make sure to correct and add that in the final version of the paper.
>
> $U^\star(s)$ is a scalar; we do not believe an infinity norm is needed. (Or maybe you meant an infinity norm over the vector $(U^\star(s))_{s\in[t]}$ ?)
>
> The novelty of our work is to target the exponential term in the regret of [18] and mainly their step 4 that we replaced completely by intersecting the event $\mathfrak{G}$ ($\mathfrak{C}$ in their notation) with the event of the clean run $\mathfrak{A}$ that we introduced.
>
> Our proof relies on a maximal concentration inequality in the same spirit as Lemma 3 of  [9] to bound the maximal deviation of the estimate of the reward of the best action (event $\mathfrak{D}$). This result was not used in [18] and is a key element in our proof.
> Some more basic concentration inequalities are used to control the probability of event $\mathfrak{B}, \mathfrak{C}$. And for the event $\mathfrak{F}$ like in [18]. The way the event $\bar{\mathfrak{H}}$ is handled is not new and is already handled by the literature [18], or [9] and now (https://arxiv.org/pdf/2302.11182).
>
>
> We are not convinced that this technique of added exploration could be used in other reward settings to alleviate the exponential regret. Even for 1-Lipshitz reward functions, the task seems rather tedious. First, we would need to find a way to extend the definition of $f$ to $\mathbb{R}^d$ and not $[a,b]^d$. For instance, the example of theorem 6 of [22] $f$ is Lipshitz on $[0,1]^d$ but not on $\mathbb{R}^d$. Furthermore, suppose one tries to apply SG-CTS to that example. In that case, we believe that in order to explore the optimal decision, all the Thompson samples would need to be simultaneously greater than a certain threshold. This is a very low probability set in high dimensions. We believe that, in general, the structure of Lipshitz functions could induce a lot of confusing sets of parameters where the initial prior would put a very small probability. If that were to happen, Thompson sampling would need an exponentially long time to explore all those zones. It is what happens in step 4 of [18]. Our method works for linear functions because there are essentially only two sets to be explored, and our posterior still puts enough probability on both of them.

---

> > ### Comment · Reviewer_yJpu · 2024-08-13
> >
> > I read the authors’ rebuttal and skimmed the comments of other reviewers. I would like to keep my score unchanged.

---

### Official Review · Reviewer_4UPn · 2024-07-03

**Soundness:** 3
**Presentation:** 3
**Contribution:** 3
**Rating:** 6
**Confidence:** 3

**Summary:**

This paper addresses Thompson Sampling for stochastic combinatorial bandits with sub-Gaussian rewards. In this area, previous work has identified the interesting phenomenon that some versions of Thompson sampling incur a per-instance regret which is exponential in the (maximum) number of arms pulled per round, m. This is not experienced by UCB-style algorithms, although those tend to have a worse computational complexity.

This paper furthers the discussion around the performance of TS in combinatorial bandits by identifying a variant which incurs only polynomial dependence on m. This is achieved by using a distribution with inflated variance to draw Thompson samples, rather than the 'true' posterior.

The paper contains non-trivial theoretical work to derive this bound, and contains an experiment demonstrating the superiority of the new TS approach over 'natural' variants and its comparable performance to ESCB, a state-of-the-art UCB-based algorithm.

**Strengths:**

I find the research questions considered in this paper very interesting, and think that the multi-armed bandits community at NeurIPS will value this work. Thompson Sampling is a very popular algorithm for all kinds of bandit problems, and the fact that its most natural extension to combinatorial bandits has poor performance is an important issue. The theoretical results in this paper meaningfully contribute to the understanding of where 'natural' Thompson sampling is necessary and where it is only 'some notion of adding randomness' that is needed.

The theoretical work is based on state-of-the-art tools, does include some novel steps and is likely to be of its own interest. It does not appear to be a routine translation of existing tools.

The paper is mostly well-written and clear, and for someone acquainted with combinatorial bandits it is quite easy to follow.

**Weaknesses:**

The main weakness of the paper is that more could be done to articulate exactly where the results improve over existing bounds. The exponential and polynomial dependence on m is, as the paper states, in the constant order (wrt T) term. The main comparison between the results of the present paper and [18] is in terms of order results, and some constants are potentially large. It leaves the reader uncertain as to where in the space of T, m and d, the bound of this paper improves upon the bound of [18]. Adding some clarity around this would improve the paper, and help to establish the extent of its contribution.

There are some further points where I feel clarification could be made around theoretical and experimental results. I ask questions about these in the section below, and they are also central to me assessing the extent of the contribution.

There are some parts where the explanations are probably a bit too brief to be accessible to those without expert knowledge of the field - e.g. lines 52-64 discussing related literature use undefined technical terminology and discuss papers very quickly.

**Questions:**

1. For which values of m and T specifically does the bound of this paper improve upon the bound in [18]? I appreciate this is problem dependent, but if this could at least be answered for the example considered in the experiments, that would be beneficial.

2. The situation in the experiments seems as though it could also be tackled by a 2-armed bandit policy that receives full-bandit feedback on [0,d/2] and essentially ignores the combinatorial structure. If those rewards were scaled to [0,1] would the combinatorial algorithms be outperformed by algorithms for 2-armed bandits?

3. When introducing B-CTS and G-CTS could you make clear if these are the variants to which the results of [18] and [22] apply?

4. The conclusion that using a mismatched variance term outperforms the natural variance is an interesting one, and seems to mirror some findings in the bandit literature that randomisation strategies that are not TS-like or based on a posterior distribution (explicitly) can perform well in bandit problems (e.g. Perturbed History Exploration and bootstrapping based approaches, Kveton et al. (2019) and subsequent work). Can you comment on whether this polynomial dependence relies on using something close to the posterior, or whether other randomisation strategies would seem to be sensible candidates?

5. Can you add to section 5 some clarity on what steps are novel and what are inspired by previous theoretical work? Some of the construction of a clear run seems to bear resemblance to (now) classic proofs for Thompson sampling (Agrawal and Goyal (2012), Kaufmann et al (2012), etc.) and the idea of using sample paths may have roots in other work too? (I'm not certain on that)



Agrawal and Goyal (2012) http://proceedings.mlr.press/v23/agrawal12/agrawal12.pdf
Kaufmann et al (2012) https://arxiv.org/pdf/1205.4217
Kveton, Szepesvari, et al. (2019) https://arxiv.org/abs/1902.10089

**Limitations:**

yes

---

> ### Author Rebuttal · Authors · 2024-08-06
>
> 1.
>     + From a theoretical point of view :
> We share the same leading term in $\log{T}$. However, for the exponential [18] vs. polynomial term, you need to compare $m^{8m}$ and $m^{20}  \times d^{10}$. So, one can say that for $m>10$, our bound can be better. This does not take into account the term in $\frac{1}{\Delta_{\min}}$. To be sure, if $m>20$, our bound is better.
>     + From an experimental point of view :
> There is no way to reach the asymptotic bound in practice (for some parameter) of [18] because the logarithmic term in $T$ is too small compared to the exponential term. In theory, it is also impossible to reach our bound because $\log(T)$ is very small compared to our polynomial term and for all practical $T$. However, it happens that in the simulation, the asymptotic regime is reached much faster than the theory predicts. Our simulations run for $m$ in a range $[5,65]$. One can already see the exponential gap happening around $m=13$.
>
> 2. Yes. This is undead the case! However, one could imagine a slightly more complex set where one could not easily separate the arms into separate actions. (A set with many decisions with many arms but which share a few of them.) This would also create exponential regret in the Semi-bandit setting.
>
> 3. For B-CTS and G-CTS, this is the version that [18] and [22] apply in the linear setting. However, in [18], they also provide a G-CTS with a slight modification to handle correlated subgaussian noise, but it still shows exponential regret.
>
> 4. If we understand the reviewer correctly, we believe that the randomization that Kveton et al. (2019) use could indeed help an algorithm like Thompson sampling (B-CTS) to work a bit better without alleviating the exponential regret. In fact, those constant terms (polynomial or exponential) come from a kind of waiting time for the best action to be played enough time. As far as we know, randomization tries to lower the probability that the reward of the best action is underestimated. However, if the best action is not played enough, like in the two action examples, this would not help much. This randomization could help, for instance (in our case), so that the events $\mathfrak{D}$ and $\mathfrak{F}$ happen with a smaller probability or that we can have tighter bounds. This is indeed very interesting!
>
> 5. The idea that, to control the behaviour of TS, one should lower the bound on the number of times the optimal decision is selected is undoubtedly not new. It was indeed exploited in Agrawal and Goyal (2012), Kaufmann et al (2012) for the multi-armed bandit setting. We also believe those ideas inspired [18], [21] in their proofs for the combinatorial setting. However, some more intricate proof techniques required to handle the exponential term are new.

---

> > ### Comment · Reviewer_4UPn · 2024-08-10
> > **Response to Rebuttal**
> >
> > Thanks very much for your response to my comments, all of these responses are convincing, I would just like to know if and how/where you intend to incorporate these responses into the updated paper?

---

> > > ### Author Response · Authors · 2024-08-12
> > >
> > > Yes, we will incorporate those responses into our paper. We will emphasise the heuristic of the proof and especially the comparison with [18], Agrawal and Goyal (2012), Kaufmann et al (2012). We will also try to incorporate the exponential vs the polynomial scaling if the space constraint allows us.

---

> > > > ### Comment · Reviewer_4UPn · 2024-08-13
> > > >
> > > > Thank you, I will in that case raise my score, I'd encourage that even if the space prohibits including in the main text some remark should be included in an appendix, or some reorganisation done to allow this inclusion.

---

### Official Review · Reviewer_npKN · 2024-07-07

**Soundness:** 4
**Presentation:** 3
**Contribution:** 2
**Rating:** 6
**Confidence:** 3

**Summary:**

The authors present a new Thompson Sampling algorithm for linear combinatorial stochastic semi-bandits. The algorithm provably achieves a better finite-time regret than previous works, and specifically without an exponential dependency on the dimension of the problem.  The authors also present a "paradox" that shows using posterior knowledge is not always beneficial.

**Strengths:**

* The paper is presented clearly and the theory is sound.
* The experiments suggest the algorithm is useful in practice.
* The presented paradox is original and interesting

**Weaknesses:**

* The main result improves the known regret only for the term that does not depend on the time horizon, which is usually less interesting.
* Code is not provided for the numerical experiments.

**Questions:**

* In line 26, do you mean $X(t)$ is uniformly distributed?
* Can you provide summary of the algorithms as figures? It is hard to follow in the text what is the exact algorithm
* Can you provide any intuitive explanation for the presented "paradox"?

**Limitations:**

The authors address the limitations of the paper.

---

> ### Author Rebuttal · Authors · 2024-08-06
>
> We will provide the code in an open-source repository after the review process.
>
> * In line 26, $X(t)$ could be any random variable that is bounded in $[a,b]^{d}$. However, the one that maximizes its variance is the half Dirac in $a$ and $b$. The latter gives us the most deconcentrated random variable with a subgaussian parameter $(b-a)^2/4$. And any bounded random variable is subgaussian with parameter $(b-a)^2/4$.
>
> * Due to space constraints, the algorithms as figures were removed. They will be added back to the final version of the paper. You can find it in the pdf attached to the general answer.
>
> * One of the reasons behind the paradox is the fact that if the unknown parameter $\theta$ lies in $[a,b]^d$ and we attempt to use TS to select decisions by assigning (for instance) a uniform prior over that set, then in many cases the algorithm will put too little probability mass on the set around the actual parameter $\theta$ so that TS will never explore enough to discover $\theta$ and will be stuck in an infinite loop of sampling suboptimal decisions. Indeed, when $d$ is large, any region that only contains vectors $\theta$ such that $(1/d) \sum_{i \in [d]} \theta_i$ is not close to $(b-a)/2$ will be assigned very little probability mass, from the law of large numbers. In the counter example of 2 actions, if the worst action is sampled first and gives a reward that is greater than the mean of the sum of the priors of the other action. Then, because of the prior, Thompson sampling will firmly believe it is the optimal action (again due to concentration). Therefore, it will not explore the other action, which is the best, for a very long time. [22] exploited this to create the example (see their section 3.2).

---

> > ### Comment · Reviewer_npKN · 2024-08-09
> >
> > Thank you for the response, I will keep my score

---

### Official Review · Reviewer_PpKq · 2024-07-10

**Soundness:** 3
**Presentation:** 3
**Contribution:** 3
**Rating:** 5
**Confidence:** 3

**Summary:**

This paper proposes a modified version of posterior sampling that achieves optimal asymptotic regret bound, providing an algorithm through the methodology of Thompson Sampling that achieves such a bound.

**Strengths:**

This is a technical paper, and the message is clear and intriguing. This paper validates the methodology of Thompson Sampling in the context of combinatorial bandits in terms of achieving polynomial dependence on the number of dimensions. The technique is novel, to my knowledge.

**Weaknesses:**

I am exactly doing this line of research, so I am not able to follow all the proof details. The paper is notation-heavy; for instance, there are nine events defined, eight of which are some sort of deviations. It is hard to parse and keep track of for a person outside the exact line of research.  I appreciate the authors' effort to explain, but I do believe in the main paper, the authors should write a more sketchy proof to highlight the key technique to prove novelty than writing a semi-rigorous proof: for instance, out of the eight deviation events, which are standard and bounded by conventional results? which are novel contributions? Of all the techniques involved, which is the key to this new result? In all, I feel it would be of great help if a clear logic line for the proof is demonstrated for people outside this line of research. At the current state, it is rather hard for me to evaluate the technical contribution.

**Questions:**

What is the guarantee for G-CTS?

**Limitations:**

adequately addressed

---

> ### Author Rebuttal · Authors · 2024-08-06
>
> The decomposition under a clean run (event $\mathfrak{A}$) is original. The decomposition of event $\mathfrak{A}$  into  events $\mathfrak{B},\mathfrak{C},\mathfrak{D}, \mathfrak{E}$  is new. And how we handle event $\mathfrak{E}$ is original. However, the handling of event  $\mathfrak{B},\mathfrak{C},\mathfrak{D}$ are similar to prior work. The event $\mathfrak{Z}, \mathfrak{G}, \mathfrak{F}, \mathfrak{H}$ are not new and comes from [18]. However the way we handle $\bar{\mathfrak{G}} \cap \mathfrak{A}$ is.
>
> The regret upper bound for G-CTS (Not SG-CTS) is given in the paper [18] Theorem 3. One needs to go to the appendices D, page 21, to see the exponential term. It states that under linear rewards, subgaussian noise of the rewards of the items (note that the noise can be correlated but known), the regret scale in $\frac{d\ln(m)^{2}\ln(T)}{\Delta_{\min}}$. However, there is an exponential constant term. This is why we decided to add a little more exploration and to rework the analysis.

---

> > ### Comment · Reviewer_PpKq · 2024-08-13
> > **Reply to rebuttal**
> >
> > The authors have addressed my concern. I would like to maintain my score.

---

### Author Rebuttal · Authors · 2024-08-06

We want to thank the reviewers for their questions and remarks; we will take them into account to improve the paper's clarity. Here is a general answer to all the questions asked by the reviewers.

The main contribution of our paper was to find a way to circumvent the exponential term in the work of [18] in step 4 of his analysis, page 13. We replace it with a polynomial term in Theorem 1. The exponential dependency is due to the fact that they control some of the regret by the expected time necessary for **all** of the Thompson samples to be simultaneously greater than a threshold, and this expected time scales exponentially in the ambient dimension. That in high dimension is an exponentially small set in probability. In fact, for the best action to be played, we only need the mean of the Thomspon samples to be greater than a certain threshold. Furthermore, we needed to add more variance to them to control some deviation of the estimate of the rewards.

Below are some of the original proof elements we came up with to circumvent the problem:

1. First, we show that what we name a clean run happens with a high probability. Proposition 2. The way we decompose ${A^{\star}}^\top \theta(s)$ is original with the introduction of the random part of the Thompson sample $Z(t)$. The treatment of the event $\mathfrak{E}$ is original. It is a mix between a deconcentration inequality (i.e. a lower bound on the probability that a random variable is far from its expectation), a study of the function $f(t)/g(t)$, and a multiplicative Azuma Chernoff bound. The multiplicative Azuma Chernoff bound is, we think, classic, but we did not manage to find a reference for the direction of inequality, which we proved in the appendix.

2. Second, under the clean run event, we show that the optimal action is played at least a certain number of times. This is done in Proposition 3. It combines a concentration result from [9] and our last result.

3. Last, The subsubsection 5.3.2 is new and uses the last two results. It replaces the step 4, page 13 of [18]


It was discussed that a graph representing the event and the regret decomposition, highlighting the novel contribution, would be added to the paper's appendices. If the reviewers believe that it facilitates the understanding of the proof, we propose adding it to the paper's final version.

We hope this answer will help the reviewers understand our work's novelty. Please do not hesitate to reach out for more details if needed.

Here are the algorithms in figure form.

---

### Decision · Program_Chairs · 2024-09-25

**Decision:**

Accept (spotlight)

**Comment:**

The paper addresses a core problem in combinatorial multi-armed bandit algorithms -- of controlling the scaling of regret with the problem size. It contributes a carefully designed variant of Thompson sampling (TS) that alleviates the known exponential scaling of naive TS' regret with the number of arms. It also raises an interesting paradox about when TS is allowed to use mismatched likelihood distributions and can perform better than the vanilla one.

All the reviewers (and I) agree that the paper makes solid contributions to bandit algorithm design; thus I recommend the paper for acceptance.